

# Patterns of microbial processes shaped by parent material and soil depth in tropical rainforest soils

**Laurent K. Kidinda[1,2], Folasade K. Olagoke[1], Cordula Vogel[1], Karsten Kalbitz[1], Sebastian Doetterl[3,4]**

[1]Chair of Soil Resources and Land Use, Institute of Soil Science and Site Ecology, TU Dresden, Germany

[2]Biogeochemistry and ecology of tropical soils and ecosystems, University of Lubumbashi, DR Congo

[3]Institute of Geography, Augsburg University, Augsburg, Germany

[4]Terrestrial Ecosystems, ETH Zurich, Zurich, Switzerland

**Correspondence :** Laurent K. Kidinda (laurent.kidinda@mailbox.tu-dresden.de)

**Abstract**

Microbial processes are one of the key factors driving carbon (C) and nutrient cycling in terrestrial ecosystems, and are strongly driven by the equilibrium between resource availability and demand. In deeply weathered tropical rainforest soils of Africa, it remains unclear whether patterns of microbial processes differ between soils developed from geochemically contrasting parent materials. Here we show that resource availability across soil depths and regions from mafic to felsic geochemistry shapes patterns of soil microbial processes. During a 120-day incubation experiment, we found that microbial biomass C and

extracellular enzyme activity were highest in the mafic region. Microbial C limitation was highest in the mixed sedimentary region and lowest in the felsic region, which we propose is related to the strength of contrasting C stabilization mechanisms and varying C quality. None of the investigated regions and soil depths showed signs of nitrogen (N) limitation for microbial processes. Microbial phosphorus (P) limitation increased with soil depth but was similar across geochemical regions, indicating that subsoils in the investigated soils were depleted in rock-derived nutrients and are therefore dependent on efficient biological

recycling of nutrients. Microbial C limitation was lowest in subsoils, indicating that subsoil microbes can significantly participate in C cycling and limit C storage if increased oxygen availability is prevalent. Using multivariable regressions, we demonstrate that microbial biomass C normalized to soil organic C content (MBC$_{SOC}$) is controlled by soil geochemistry and substrate quality, while microbial biomass C normalized to soil weight (MBC$_{Soil}$) is predominantly driven by resource distribution. We conclude that due to differences in resource availability, microbial processes in deeply weathered tropical

rainforest soils greatly vary across geochemical regions which must be considered when assessing soil microbial processes in organic matter turnover models.

**Keywords** : extracellular enzymes, microbial biomass, carbon limitation, nutrient limitation.



## 1 Introduction

Soil microbial activity is a key factor defining the fate of carbon (C) and nutrients in terrestrial ecosystems (Fang et al., 2014). Soil microbes are responsible for the release of the largest amount of $CO_2$ from soil to the atmosphere, provide nutrients for plant growth, and incorporate up to 5 % of soil organic C (SOC) in their living biomass (Jenkinson and Ladd, 1981; Luo and Zhou, 2010; Cleveland et al., 2013; Kallenbach et al., 2016). Thus, microbial biomass constitutes a biologically active fraction of the organic C pool and can act as both a source and a sink of C and nutrients (Inubushi and Acquaye, 2004).

The equilibrium between microbial resource demand and soil resource availability is one of the key factors defining patterns of microbial biomass (Allison and Vitousek, 2005; Malik et al., 2020). This equilibrium is reflected by the activity and stoichiometry of extracellular enzymes that are responsible for degrading complex organic compounds to acquire C and nutrients to meet the microbial demand (Sinsabaugh et al., 2009, Moorhead et al., 2013). Hereby, microbial communities can regulate greatly the overall production and release of extracellular enzymes mining particular nutrients in order to meet the demands of the community (Moorhead and Sinsabaugh, 2006; Doetterl et al., 2018).

When a resource is scarce, enzyme production towards its acquisition increases, whereas on the other hand, when the targeted resource is abundant enzyme production declines (Koch, 1985; Allison et al., 2011).

Broad-scale studies often report contrasting patterns of microbial biomass and enzyme activity between tropical and temperate soils as a result of varying climate and geochemical soil properties (e.g. Sinsabaugh et al., 2008; Xu et al., 2013; Margalef et al., 2017; Jing et al., 2020). In temperate soils, microbial activity driving organic matter decomposition is strongly regulated by seasonal temperature and water availability (Vitousek and Howarth, 1991). Additionally, episodic glaciations have rejuvenated land surfaces and soils which are still rich in releasable rock-derived nutrients. Thus, microbes in temperate soils are often C- and nitrogen- (N) limited (Chadwick et al., 1999; Hou et al., 2012; Chadwick and Asner, 2017). Unlike temperate soils, microbial processes in tropical rainforest soils take place nearly all year long under wet and warm conditions in soils that are high in Fe and Al (hydr-) oxides and low activity clays, while being low in pH due to long-lasting weathering processes (Chadwick and Asner, 2017). Additionally, chemical weathering in tropical systems has depleted mineral nutrients from soils, which together with P immobilization due to low pH (Baillie, 1996; Mercado et al., 2011), often results in P-limitation (e.g. Wang et al., 2019, Jing et al., 2020).

Beyond the broad generalization that characterizes soils of the humid tropics as highly weathered and acidic, with low cation exchange capacity and relatively uniform subsoil properties, there is a diversity of physico-chemical properties that reflect the heterogeneity of parent materials (Dalling et al., 2016). Tropical soils with high clay content dominated by Fe and Al oxides and low pH have a high potential to stabilize enzymes on mineral surfaces (Dove et al., 2020). Enzymes stabilized on mineral surfaces can be out of microbial regulation, thus affecting microbial C and nutrient acquisition (Allison and Vitousek, 2005; Liu et al., 2020). It is therefore unclear whether patterns in microbial processes in tropical Africa differ between soils of contrasting geochemical origin, especially those that are deeply weathered. Furthermore, physico-chemical controls on microbial processes in deeply weathered soils are still poorly understood.



Studies from subtropical and temperate regions show that microbial activity decreases with soil depth because of low C and nutrient contents in subsoils (e.g. Blume et al., 2002; Kramer et al., 2013; Stone et al., 2014; Loeppmann et al., 2016; Liu et al., 2020, Jing et al., 2020). However, low microbial activity in subsoil can also reflect a dominance of microbes that have adapted to low resource availability conditions (Hoehler and Jorgensen, 2013; Stone, 2014) and are not necessarily more C- and nutrient-limited than topsoil microbes. To date, most studies on microbial biomass, as well as enzymatic activity, are restricted to topsoils, <20 cm (e.g. Cleveland et al., 2002; Turner et al., 2010; Waring et al., 2014; Camenzind et al., 2018; Wang et al., 2019). Given that microbial C and nutrient limitations in subsoils are rarely studied (e.g. Jing et al., 2020), their patterns across geochemical gradients in deeply weathered soils remain unknown.

Hence, this study aims to answer the following questions: (i) do patterns of microbial processes in deeply weathered tropical rainforest soils reflect the geochemical gradient of parent material under similar climatic conditions? (ii) are subsoil microbes in highly weathered tropical rainforest soils more C- and nutrient-limited than their topsoil counterparts? and (iii) what are important controls on soil microbial biomass, C and nutrient limitations? To answer these questions, we investigated patterns of microbial processes and their controls along geochemical and depth gradients in tropical rainforest soils. We used soil materials from three geochemically distinct parent materials and three soil depths (0−70 cm) and measured microbial biomass C and enzyme activity (i.e. C, N, and P acquiring enzymes) at the beginning and end of a 120-day incubation experiment. In addition, we conducted a vector analysis based on ecoenzymatic stoichiometry to assess microbial C and nutrient limitations (Sinsabaugh et al., 2009; Moorhead et al., 2013; 2016). Then, we evaluated control of physico-chemical soil variables on microbial processes using dimension reduction technique and regression analysis.

## 2 Materials and methods

### 2.1 Study sites

Study sites were located in the border region between the Congo and Nile basin, along the east African rift valley system. The sites spread across the province of South Kivu in the eastern part of the Democratic Republic of the Congo, western Rwanda, and southwestern Uganda at 1300–2200 m altitude. The topography is strongly undulating with smaller plateaus and ridges and steep slopes (>60 % slope steepness) and smaller V-shaped valleys. The dominant vegetation in all forests across the region is a primary tropical mountain forest with differences in biodiversity and composition (van Breugel et al., 2015). The climate of the study area is classified as a tropical humid climate with weak monsoonal dynamics (Köppen Af-Am). Mean annual temperatures range between 15.3–19.2°C and mean annual precipitation between 1697–1924 mm. Despite the intense rainfall and steep topography reported for the region, almost no surface erosion on hillslopes is reported and measured under the dense tropical forest vegetation coverage (Wilken et al., 2021, same special issue).

Sites in South Kivu (−2.31439 °S; 28.75246 °E) are predominantly based on mafic magmatic rocks, typically mafic alkali-basalts ranging in age between 9–13 Ma. Sites in Uganda (0.46225 °N; 30.37403 °E) are situated on felsic magmatic and metamorphic rocks consisting typically of gneissic granites with an age between 1600–2500 Ma.



Sites in Rwanda (−2.463088 °S; 29.103834 °E) are situated on a mixture of sedimentary rocks of varying geochemistry consisting of alternate layers of quartz-rich sandstone, siltstone, and dark clay shists with an age between 1000−1600 Ma (Schlüter, 2006). Usually, unweathered mixed sedimentary rocks contain, on average, 1.07 ± 1.28 % geogenic, fossil C (maximum measured was 4 % geogenic C). Rock-derived nutrients (Ca, Mg, K, Na as the total reserve in releasable base cations, TRB) are markedly higher in bedrock samples of the mafic region (2.00 ± 1.00 mass %) than in the felsic (0.01 ± 0.02

mass %) and mixed sedimentary regions (0.10 ± 0.04 mass %). Similarly, total P is much higher in the mafic (P 0.37 ± 0.06 mass %) than in the felsic (P 0.01 ± 0.00 mass %) or mixed sedimentary region (P 0.02 ± 0.01 mass %). A similar pattern is observed for rock-derived Al, Fe and manganese (Mn). Potential ash deposition through the active volcanism in the region might add or change elemental concentration in soils to various degrees. However, no irregularities in soil chemistry were observed for the investigated forest sites (Doetterl et al., 2021, same special issue).

Following the WRB soil classification (2014), soils in the mafic region are described as umbric, vertic and geric Ferralsol and ferralic vertic Nitisols. Soils in the mixed sedimentary region and the felsic region are described as geric and vertic Ferralsol respectively. Soils in valley bottoms can locally show gleyic features, and be paired with fluvic Gleysols. Within the mafic region, soils are classified as clay loam, while soils in the felsic and mixed sedimentary region are classified as clay loams and sandy clay loams respectively (Doetterl et al., 2021).

**2.2 Soil sampling**

Soil samples were collected from March to June 2018 in each geochemical region following a strict catena approach where 40x40 m plots were established on four geomorphic positions (plateau/ridge, upslope, midslope and valley/foothill) in three replicates, resulting in a total of 36 plots. At one plot per topographic position, in each site, a soil profile pit was dug to 100 cm depth at the center to conduct a full soil description following WRB classification (WRB, 2014) (see Doetterl et al., 2021

for details). Each plot was subdivided into four subplots of 20x20 m from which four 1-meter soil cores were taken using a cylindrical soil corer (8.5 cm diameter). Cores from the same plot were combined into a composite and sampled at 10 cm depth increments. After sampling, field moist soils were sieved to 12 mm to get a homogenous sample that still contained the inherent aggregate structure before they were air-dried. For this study, three soil depths were selected for further analyses including topsoils (0−10 cm), shallow subsoils (30−40 cm) and deep subsoils (60–70 cm) representing a gradient in biogeochemical

properties.

**2.3 Incubation experiment**

To assess specific maximum heterotrophic respiration (see Bukombe et al., 2021 for details, same special issue) as well as microbial biomass C ($MBC_{Soil}$) and extracellular enzyme activity (EEA), a 120-day incubation experiment was carried out. For this, we used 50 g of 12 mm sieved soils weighed in a 100 ml beaker with soil moisture adjusted to 60 % of the maximum

water holding capacity, a capacity considered as the optimum water level for microbial activities (Rey et al., 2005). Samples were incubated at 20 °C as this is the temperature closest to the mean annual temperature of the study sites and sampled for $CO_2$ continuously throughout the experiment. To assess the change in microbial processes as a result of nutrient conservation and cycling, we measured $MBC_{Soil}$ and extracellular enzyme activity (EEA) at the start and end of the incubation experiment.





Measurements were done at the starting point of the incubation, i.e. 4 days after re-wetting soils, and at the endpoint after 120

days when microbial respiration stabilized at a low-level and no longer varied significantly between several time points of

incubation.

## 2.4  Soil enzyme assays

The potential activity of five extracellular enzymes produced by soil microbes was measured at the starting and endpoint of

the incubation. Targeted enzymes for C acquisition were cellobiohydrolase (CB) and β-glucosidase (BG) as the main enzymes

involved in the degradation of cellulose and related carbohydrates to acquire C (German et al., 2011). For N acquisition, we

targeted N-acetylglucosaminidase (NAG) and leucine-aminopeptidase (LAP) as the main enzymes involved in N

mineralization from chitin derived oligomers and peptides (Sinsabaugh et al., 2009). For P acquisition we targeted acid

phosphatase (AP) as this involved in mineralization of organic P into phosphate under acidic conditions (German et al., 2011).

All enzymes were measured fluorometrically in soil suspension following German et al. (2011). Briefly, the soil suspension

was prepared by sonicating 1 g of 2 mm sieved soil in 50 ml of 50 µM sodium acetate trihydrate buffer for aggregate disruption

using an ultrasonic-homogeniser (HD3100 Sonopuls, Bradlin, Germany) at an energy of 60 J ml$^{-1}$ and current power (W) of

34 J s$^{-1}$ (Marx et al., 2001). Additional 50 ml of sodium acetate was added to the homogenised suspension and stirred with a

magnetic stirrer before enzyme analyses.

Four methylumbelliferyl linked substrates, β-D-cellobioside (200 µM), β-D-glucopyranoside (200 µM), N-acetyl-β-D-

glucosaminide (200 µM), phosphate (400 µM), and L-Leucine-7-amido-4-methylcoumarin (100 µM) were used to measure

the activity of CB, BG, NAG, AP, and LAP, respectively. Assays included standards, sample, and control for soil and substrates

in four well replicates. Microplates were incubated at 30 °C for one hour and measured fluorometrically at an excitation

wavelength of 360 nm and an emission wavelength of 450 nm using a microplate reader (Synergy HTX Multi-Mode Reader,

Bio-Tek Instruments, Inc., USA). Then EEA normalized to dry soil weight (nmol g$^{-1}$ h$^{-1}$) was calculated as the difference in

fluorescence between sample and control relative to the slope and the intercept of the standard curve (German et al., 2011).

## 2.5 Microbial carbon and nutrients limitation

Microbial C and nutrient limitations were assessed using vector analysis (vector length and vector angle) on log-transformed

enzymatic ratios (Sinsabaugh et al., 2009; Moorhead et al., 2013, Chen et al., 2018, Dong et al., 2019) following Eq. (1) and

(2):

$Vector\ length = \sqrt{[ln(BG + CB)/ln(NAG + LAP)]^2 + ln(BG + CB)/lnAP)^2}$    (1)

$Vector\ angle\ (°) = Degrees\ ATAN2\ [ln(BG + CB/lnAP), ln\ (BG + CB/ln(NAG + LAP))]$    (2)

Hereby, microbial C limitation increased with vector length.  N and P limitations were indicated by vector angle with <45°

indicating N and >45° indicating P limitation (Chen et al. 2018, Cui et al., 2019, Jing et al. 2020).

## 2.6  Microbial biomass and dissolved organic carbon

Microbial biomass C (MBC$_{Soil}$) was measured by the chloroform fumigation-extraction method (Vance et al., 1987). Two

aliquots (~ 5 g each) of each sample (2 mm sieved) were taken, of which one was fumigated using ethanol-free chloroform,





while the other was kept untreated (i.e. non-fumigated). The fumigated aliquot was incubated for 24h at 25 °C, then chloroform was allowed to evaporate for 30 min under the fume hood.

Thereafter, 20 ml of 0.05M $K_2SO_4$ was added to both fumigated and non-fumigated aliquots and shaken for 60 min using an
end-over-end shaker at 35 rev min$^{-1}$. The extracts were then filtered through a Whatman filter paper (no. 42, Sigma-Aldrich, Germany). Organic carbon (OC) content in both extracts was measured using a Vario TOC cube (Elementar, Germany) in two technical replicates. $MBC_{Soil}$ was then calculated as the difference in extractable OC contents between fumigated ($OC_{fum}$) and non-fumigated ($OC_{non\text{-}fum}$) samples considering 0.45 as a factor of microbial biomass extraction efficiency ($k_{EC}$) (Beck et al. 1997) following Eq. (3):

$MBC_{Soil} = (OC_{fum} − OC_{non\text{-}fum})/kEC$                         (3)

**2.7 Biogeochemical soil parameters**

Soil texture, total elemental composition, bioavailable P (Bray-P), $pH_{KCl}$, exchangeable acidity, effective (ECEC) and potential (CEC) cation exchange capacity, as well as base saturation (BS) of both ECEC and CEC, were determined on air-dried aliquots before re-wetting for incubation (see Doetterl et al., 2021 for details on methods).

Total dissolved N (TDN), ammonium ($NH_4$-N), nitrate ($NO_3$-N), total N, SOC, and dissolved organic C (DOC) were measured at both pre- and post-incubation. DOC, TDN, $NH_4$-N, and $NO_3$-N were measured in $CaCl_2$ extracts (Rennert et al., 2007). Briefly, 25 ml 0.01 M $CaCl_2$ was added to 2.5 g soil in a 50 ml centrifuge tube (FalconTM tube). This was shaken at 2 cycles per second at 4 °C for 30 min and then placed in a pre-cooled (4 °C) centrifuge at 3200 x G for 30 min before the supernatants were collected and stored at −20 °C until measurement. Before measurements, the extracts were filtered using syringe filters
(CHROMAFIL ® PET-45/25, Polyester). DOC and TDN were measured using a Vario TOC cube (Elementar, Germany) while $NH_4$-N and $NO_3$-N were measured with a discrete autoanalyzer SEAL AQ400 (SEAL Analytical, Inc. USA).

**2.8 Statistics**

To evaluate the variation in MBC, extracellular enzyme activity (EEA), and microbial C and nutrient limitations across geochemical regions and soil depths, we performed a two-way analysis of variance (ANOVA) for pre- and post-incubation
separately (Webster, 2007). For each geochemical region, an additional two-way ANOVA was performed to compare means of microbial processes between pre- and post-incubation at all soil depths. Pairwise mean comparison and separation were conducted using the *emmeans* package in R. Before performing ANOVA, assumptions of normal distribution of residuals and homogeneity of variances were tested (Webster and Lark, 2019) using Shapiro-Wilk's test and Levene's test. In case of deviation from ANOVA assumptions (due to the natural variability of samples), we performed log, orderNorm, and square
root transformation on concerned variables using the *BestNormilize* package in R. Differences in microbial processes among geomorphic positions under undisturbed forest conditions were not statistically significant (Table A1) at $p<0.05$. Therefore, data from geomorphic positions were pooled, constituting 12 replicates per soil depth and geochemical region.

Controls on microbial biomass C per soil weight ($MBC_{Soil}$), microbial biomass C per SOC ($MBC_{SOC}$), C and nutrient limitations were assessed at pre- and post-incubation using dimension reduction technique and multiple linear regression models in two
steps.



First, rotated principal component analysis (rPCA) was performed on soil physico-chemical variables (measured at pre-incubation) using the *FactoMiner* package and rotated using the varimax function in R. Selected variables are those related to resource availability (e.g. organic matter) for microbial activity and those susceptible to affect resource availability and microbial living environment (e.g. pH, texture, total elements). The rPCA allowed to reduce the number of variables in the

statistical data set and to visualize the multivariate data as a set of coordinates in a high-dimensional data space. Therefore, the effects of physico-chemical soil properties, their interactions and soil depth were condensed into components with clearly distinguishable loadings. A threshold of r>0.5 was chosen to identify whether a variable contributes to the loading of a rotated component (RC) or not. Secondly, RCs with eigenvalues >1 and explained variance >5 % in addition to soil depth were used to predict $MBC_{Soil}$, $MBC_{SOC}$, C and nutrient limitations using the *caret* package (Kuhn, 2008).

The relative importance of predictors was assessed based on the *lmg* metric using the *relaimpo* package. The *lmg* approach consists of decomposing $R^2$ into non-negative contributions of each regressor (Grömping, 2006). To assess whatever soil depth mediates correlations between physico-chemical predictors and dependent variables, we conducted a partial correlation in SPSS by controlling the effect of soil depth. All statistical analyses were conducted using R software, version 4.0.1 (The-R-Core-Team, 2016) and SPSS version 25.0 (Armonk, NY, IBM Corp, 2017) considering 0.05 as the level of significance.

**3 Results**

**3.1 Patterns of microbial biomass and nutrients with soil geochemistry and depth**

Across geochemical regions, $MBC_{Soil}$ and DOC declined significantly with soil depth (p<0.05) at both pre- and post-incubation (Fig. 1). The $MBC_{Soil}$ was highest in the mafic region in top- and subsoils. Lowest $MBC_{Soil}$ was found in topsoils of the mixed sedimentary region and in subsoils of the felsic region at both pre- and post-incubation. The DOC was highest in the mafic

region and lowest in the felsic region for both top- and subsoil consistently for both post- and pre-incubation. In contrast, $MBC_{SOC}$ and $MBC_{DOC}$ generally showed no significant differences with soil depth for mafic and mixed sedimentary regions. Increased $MBC_{SOC}$ and $MBC_{DOC}$ with soil depth were determined in the felsic region at pre-incubation. The $MBC_{DOC}$ and $MBC_{SOC}$ were highest in the felsic region and lowest in the mafic region (Fig. 1l and Fig. 1j). At post-incubation, $MBC_{SOC}$ and $MBC_{DOC}$ decreased with soil depth for all regions. The $MBC_{DOC}$ was lowest in the mixed sedimentary region, while it was

slightly higher in mafic and felsic regions.





**Figure 1.** Patterns of microbial biomass C (MBC) and dissolved organic C (DOC) across geochemical regions and soil depths. Error bars are standard errors of means (n=12). Lower-case letters compare geochemical regions per soil depth at pre-incubation. Upper-case letters and asterisks (*) compare geochemical regions per soil depth at post-incubation.


Across geochemical regions, TDN, NH$_4$-N, and NO$_3$-N declined significantly with soil depth (p<0.05) for both pre- and post-incubation (Fig. 2). Highest values were observed at post-incubation, except for NH$_4$-N in the felsic region. The TDN and NH$_4$-N were highest in the mafic region for both pre- and post-incubation. The NO$_3$-N, however, was highest in the mixed sedimentary region at pre-incubation and in the mafic region at post-incubation.



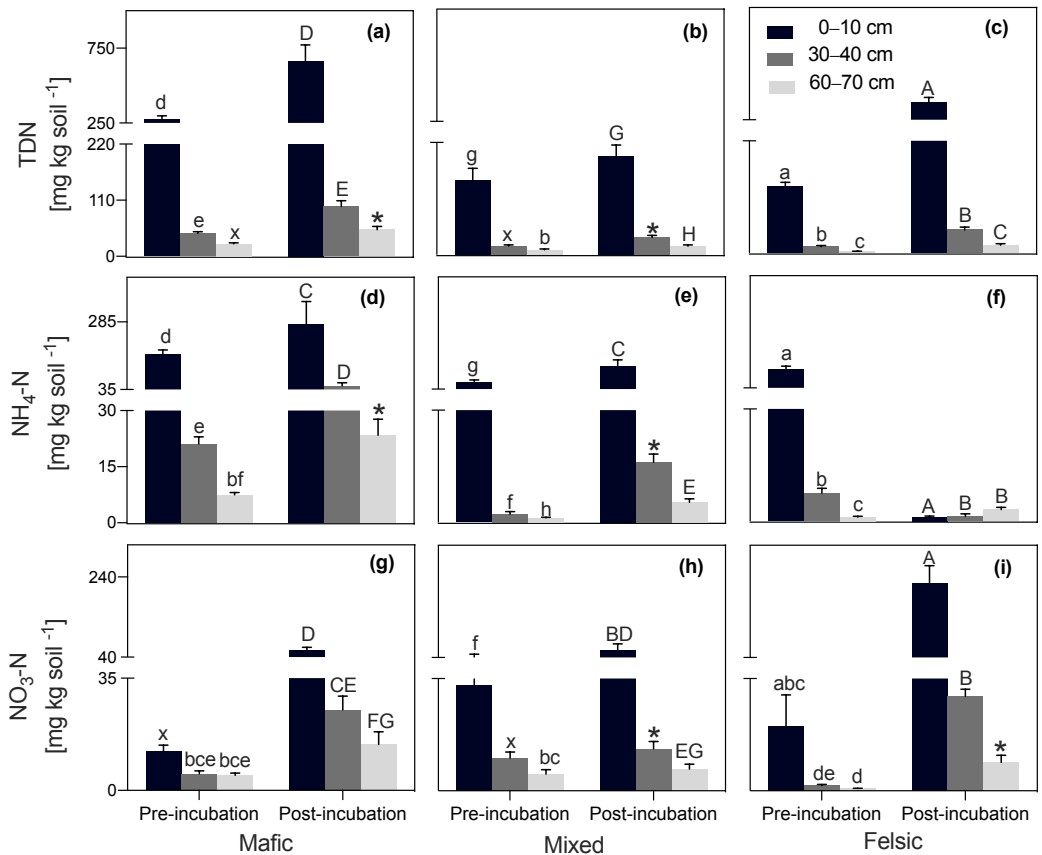

**Figure 2.** Patterns of total dissolved N (TDN) and mineral N across geochemical regions and soil depths. Error bars are standard errors of means (n=12). Lower-case letters compare geochemical regions per soil depth at pre-incubation. Upper-case letters and asterisks (*) compare geochemical regions per soil depth at post-incubation.

### 3.2 Patterns of extracellular enzyme activity with soil geochemistry and depth

In general, C, N, and P acquiring enzymes declined significantly with soil depth ($p < 0.05$) for both pre- and post-incubation and were highest in the mafic region and lowest in the felsic region (Fig. 3). The activity of all enzymes declined significantly at post-incubation, usually up to 3-fold in topsoils and 13-fold in subsoils. For each geochemical region, P acquiring enzymes were the most active, while N acquiring enzymes were the least active. There was no significant difference in the activity of P acquiring enzymes between the mafic and mixed sedimentary regions at all soil depths ($p > 0.05$). Similarly, no significant difference in N acquiring enzyme activity was found between the mixed sedimentary and felsic regions.



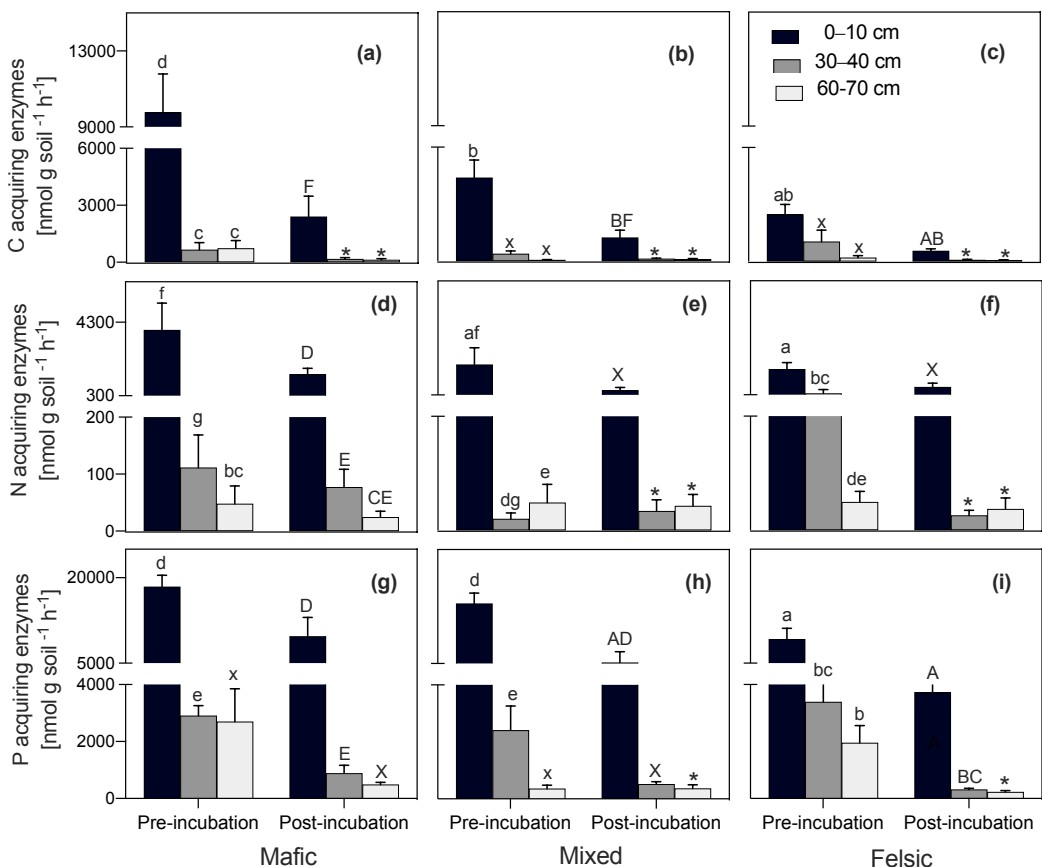

**Figure 3.** Patterns of extracellular enzyme activity across geochemical regions and soil depths. Error bars are standard errors of means (n=12). Lower-case letters compare geochemical regions per soil depth at pre-incubation. Upper-case letters and asterisks (*) compare geochemical regions per soil depth at post-incubation.

### 3.3. Patterns of nutrient limitations with soil geochemistry and depth

Across all geochemical regions, the vector angle was >45° at all soil depths and indicated microbial P limitation rather than N limitation (Fig. 4 a–c). The vector angle was similar ($p > 0.05$) among geochemical regions but was higher in sub- than topsoils ($p < 0.05$) at both pre- and post-incubation. At pre-incubation, vector length indicating microbial C limitation was highest in the mixed sedimentary region and lowest in the felsic region for both top- and subsoils ($p < 0.05$). At post-incubation, microbial C limitation declined in topsoils and was similar among geochemical regions and soil depths ($p > 0.05$).





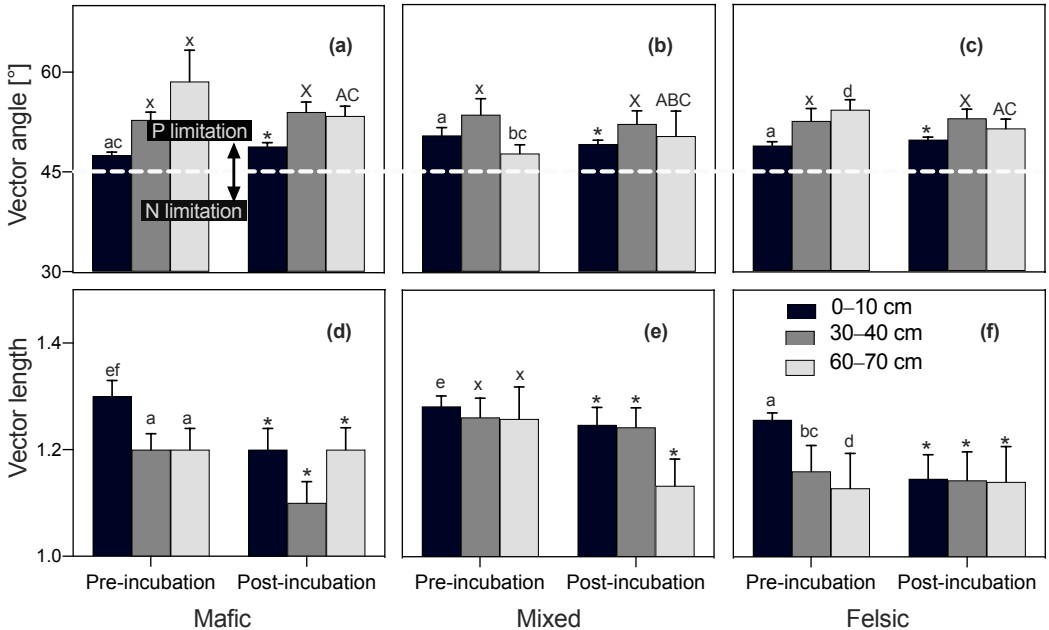

**Figure 4.** Vector analysis of microbial C and nutrient limitations. Error bars are standard errors of means (n=12). Vector length increases with microbial C limitation. Vector angle with <45° indicates N and >45° indicates P limitation. Lower-case letters compare geochemical regions per soil depth at pre-incubation. Upper-case letters and asterisks (*) compare geochemical regions per soil depth at post-incubation.

## 3.4 Rotated components and mechanistic interpretation

The rotated principal component analysis resulted in six components explaining 80.5 % of the total variance of soil properties across all three regions and soil depths (Table 1). The component RC3 explaining the highest variance (23 %) was loaded predominantly by variables that are mechanistically related to soil organic matter. The component RC1 (21.6 %) was predominantly characterized by solid acidity and exchangeable cations. Whereas, the component RC2 (20 %) mechanistically reflected the soil phase geochemistry. Components RC4 and RC5 reflected the substrate quality and nitrate content, respectively.



**Table 1.** Rotated components using physicochemical variables from all studied geologies and soil depths. SOC: soil organic C, DOC: dissolved organic C, TDN: total dissolved N, CEC: cation exchange capacity.

|  |  |  | RC3 | RC1 | RC2 | RC4 | RC5 |
|---|---|---|---|---|---|---|---|
| Eigenvalues |  |  | 6.21 | 5.83 | 5.40 | 2.39 | 1.89 |
| Proportion variance (%) |  |  | 23 | 21.6 | 20 | 8.8 | 7 |
| Cumulative variance (%) |  |  | 23 | 44.6 | 64.6 | 73.5 | 80.5 |
|  | **Mechanistic interpretation** | Unit | **Organic matter** | **Acidity and exchangeable cations** | **Solid phase geochemistry** | **Substrate quality** | **Nitrate** |
| Organic matter and soluble nutrients | Total N | mg kg$^{-1}$ | **0.96** | 0.11 | 0.09 | 0.1 | -0.01 |
|  | SOC | mg kg$^{-1}$ | **0.95** | -0.11 | -0.07 | -0.16 | -0.05 |
|  | Soil C/N |  | -0.1 | -0.14 | -0.11 | **-0.85** | 0.06 |
|  | NH$_4$-N | mg kg$^{-1}$ | **0.89** | 0.28 | -0.01 | 0.05 | 0.08 |
|  | NO$_3$-N | mg kg$^{-1}$ | 0.36 | -0.06 | -0.18 | 0.01 | **-0.72** |
|  | TDN | mg kg$^{-1}$ | **0.93** | 0.11 | -0.04 | 0.07 | -0.09 |
|  | DOC | mg kg$^{-1}$ | **0.8** | -0.36 | 0.03 | 0.11 | 0.2 |
|  | DOC/SOC |  | 0.02 | -0.45 | 0.19 | 0.42 | 0.32 |
|  | NH$_4$-N/TDN |  | 0.44 | **0.59** | -0.05 | 0.07 | 0.32 |
|  | NO$_3$-N/TDN |  | -0.31 | -0.21 | 0.06 | -0.17 | **-0.75** |
|  | DOC/(NO$_3$-N+NH$_4$) |  | -0.34 | -0.38 | -0.21 | -0.24 | 0.45 |
|  | Bio-available P | mg kg$^{-1}$ | 0.32 | 0.11 | -0.22 | -0.06 | 0.47 |
| Acidity and exchangeable cations | pH |  | -0.13 | **0.93** | -0.07 | 0.06 | 0.05 |
|  | Effective CEC | me 100g$^{-1}$ | **0.53** | **0.73** | 0.03 | 0.13 | -0.01 |
|  | CEC | me 100g$^{-1}$ | 0.7 | -0.03 | **0.51** | 0.23 | 0.08 |
|  | Base saturation (BS) | me 100g$^{-1}$ | -0.05 | **0.9** | -0.29 | 0.14 | 0.02 |
|  | Exchangeable acidity | me 100g$^{-1}$ | 0.4 | **-0.78** | 0.19 | 0.03 | -0.14 |
|  | Exchangeable bases | me 100g$^{-1}$ | 0.29 | **0.9** | -0.05 | 0.1 | 0.04 |
|  | BS/Clay |  | -0.03 | **0.8** | -0.43 | 0.13 | 0.04 |
| Texture | Clay | mass % | -0.15 | -0.08 | **0.91** | 0.14 | -0.06 |
|  | Silt | mass % | -0.02 | -0.2 | 0.07 | -0.89 | -0.04 |
|  | Sand | mass % | 0.15 | 0.18 | **-0.86** | 0.34 | 0.08 |
|  | Clay/Sand |  | -0.16 | -0.23 | **0.86** | -0.23 | -0.01 |
| Total elements | Total P | mass % | 0.42 | 0.03 | **0.75** | 0.15 | 0.05 |
|  | Total Al | mass % | 0.13 | -0.06 | **0.87** | 0.34 | 0.11 |
|  | Total Fe | mass % | 0.14 | -0.31 | **0.88** | 0.06 | -0.16 |
|  | Total Mn | mass % | 0.39 | **0.57** | 0.48 | 0.33 | 0.18 |





### 3.5 Controls on microbial biomass and nutrient limitations

Soil depth together with physico-chemical properties explained significantly (p=0.00) the variance in $MBC_{Soil}$, C and P limitations at pre- and post-incubation and only at post-incubation for $MBC_{SOC}$ (Fig. 5). Explained variance in $MBC_{Soil}$ ($R^2$= 0.44, RMSE=1.33) and $MBC_{SOC}$ ($R^2$= 0.10, RMSE=1.38) at pre-incubation increased by 0.3- and 3-fold at post-incubation,

respectively. In contrast, explained variance in C ($R^2$=0.19, RMSE=1.36) and P ($R^2$= 0.30, RMSE=1.34) limitations at pre-incubation decreased by 0.9- and 1.5-fold at post-incubation, respectively.

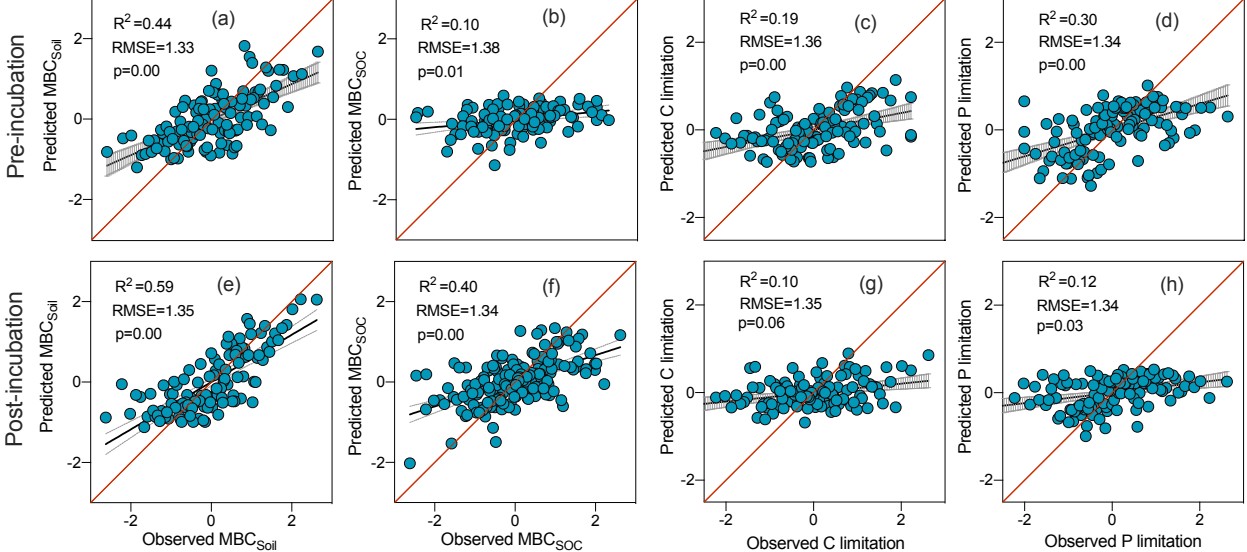

**Figure 5.** Linear regression between observed and predicted $MBC_{Soil}$ (microbial biomass C per soil weight), $MBC_{SOC}$ (microbial biomass per SOC), and CP limitations (n=108).

Organic matter and soil depth were the most important controls for $MBC_{Soil}$ at pre- and post-incubation (Table 2). Organic matter and soil depth explained 46 % and 36 % respectively of the variance in $MBC_{Soil}$ at pre-incubation, while at post-incubation 49 % was explained by organic matter and 37 % by soil depth (p<0.05). In contrast, $MBC_{SOC}$ was mainly driven by acidity and exchangeable cations (28 % of explained variance), substrate quality (27 %), and solid phase geochemistry (13 %) at post-incubation (p<0.05). The C limitation was mainly controlled by soil depth (45 % of explained variance) and organic

matter (28 %) at pre-incubation and by acidity and exchangeable cations (41 %) at post-incubation (p<0.05). The P limitation was significantly (p<0.05) controlled by organic matter (37 % of explained variance) at pre-incubation.



**Table 2.** Relative importance (%) of explanatory variables. $MBC_{Soil}$: microbial biomass C per soil weight, $MBC_{SOC}$: microbial biomass per SOC. Empty cells = non-significant relative importance ($p<0.05$).

| Explanatory variables | Pre-incubation | | | | Post-incubation | | | |
|---|---|---|---|---|---|---|---|---|
| | $MBC_{Soil}$ | $MBC_{SOC}$ | C limitation | P limitation | $MBC_{Soil}$ | $MBC_{SOC}$ | C limitation | P limitation |
| Organic matter | 45.6 | | 29.2 | 37.3 | 49.3 | | | |
| Soil depth | 35.7 | | 44.8 | | 38.6 | | | |
| Acidity and exchangeable cations | | | | | | 28.5 | 41.1 | |
| Substrate quality | | | | | | 26.6 | | |
| Solid phase geochemistry | | | | | | 12.7 | | |
| Nitrate | | | | | | | | |
| $R^2$ | 0.44 | 0.10 | 0.19 | 0.30 | 0.59 | 0.40 | 0.10 | 0.12 |
| RMSE | 1.33 | 1.38 | 1.36 | 1.34 | 1.35 | 1.34 | 1.35 | 1.34 |
| Observations | 108 | 108 | 108 | 108 | 108 | 108 | 108 | 108 |

When controlling for soil depth, correlation of organic matter with $MBC_{Soil}$ declined at both pre- (from r=0.62 to r=0.39) and post-incubation (from r=0.73 to r=0.48), but remained significant (p<0.05) (Fig. 6). None of the explanatory variables were significantly correlated to $MBC_{SOC}$ after controlling for soil depth at pre-incubation. At post-incubation, however, acidity and exchangeable cations, solid phase geochemistry, and substrate quality, remained significantly correlated to $MBC_{SOC}$. When controlling for soil depth, correlations between organic matter and C limitation disappeared at pre-incubation (Fig. 6). However, the correlation of acidity and exchangeable cations as well as substrate quality with C limitation increased significantly. The correlation of organic matter with P limitation at pre-incubation declined (from r=-0.44 to r=-0.21, p<0.05) and disappeared at post-incubation (from r=-0.27 to r=-0.09), while controlling for soil depth. Similarly, correlation between acidity and exchangeable cations with P limitation disappeared after controlling for soil depth at pre-incubation.




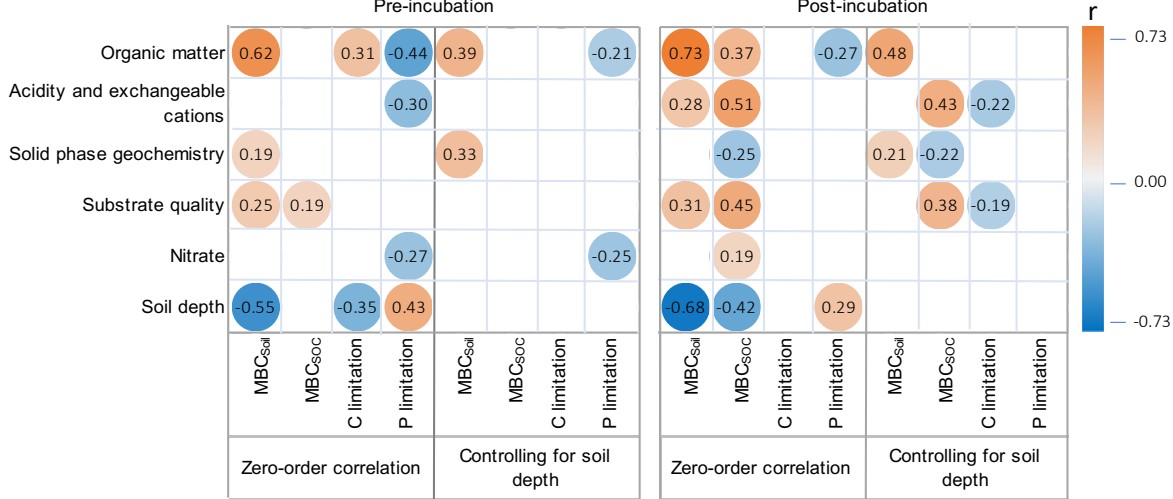

**Figure 6.** Partial correlations (controlling for soil depth) between explanatory variables and dependent variables compared to zero-order correlations. Only significant variables shown (Empty cells = non-significant correlations) (p<0.05). MBC$_{Soil}$: microbial biomass C per soil weight, MBC$_{SOC}$: microbial biomass per SOC.

## 4 Discussion

### 4.1 Patterns of microbial processes among soils of contrasting geochemical origin differ due to differences in resource availability

Microbial biomass C (MBC$_{Soil}$) and EAA followed the gradient in resource abundance, particularly C and N contents (Fig. 1−3). Differences in C and N content among geochemical regions could be due to differences in organic matter input and their interactions with geochemical soil properties regulating its stabilization. The highest C and N contents along with the highest MBC$_{Soil}$ and EAA were found in the mafic region. Highest C stabilization potential, favoured by highest pedogenic oxides has been reported by Reichenbach et al. (2021, same special issue) as a strong driver of C stock in the mafic region. In addition to C content, C quality could also affect patterns of microbial processes. For instance, the mixed sedimentary region showed over 2 times more DOC than the felsic region, however over 0.5 times less MBC$_{Soil}$ in topsoils (Fig. 1 a–f). This discrepancy indicates a different C quality between the mixed sedimentary and felsic region. In the mixed sedimentary region, Reichenbach et al. (2021, same special issue) reported up to 52 % of geogenic organic C, which is poorly accessible to microbes due to its chemical composition. Although up to 39 % of the respired C was of geogenic origin (Bukombe et al., 2021), its decomposition requires specialized microbes (Cohen and Gray, 1993), hence the reason for lower MBC$_{Soil}$ and EEA in this site. The lowest C availability in the mixed sedimentary region is supported by observed lowest MBC$_{DOC}$ and MBC$_{SOC}$ (Fig. 1). Indeed, small amounts of MBC within a large pool of C means that the average availability of C sources must be low (Insam and Domsch, 1988). Consequently, microbial C limitation was significantly higher in the mixed sedimentary region compared to the mafic (particularly in subsoils) and felsic regions (Fig. 4).





We found that soil microbes were P-limited rather than N-limited throughout all three regions. This can be attributed to

weathering losses of P-containing primary minerals and P occlusion by secondary minerals (Chadwick et al., 1999). Also, due to a long-lasting efficient recycling of organic matter, tropical forest soils are rarely N-limited (Chadwick and Asner, 2017). Although this finding is not new; it corroborates several studies in tropical and subtropical forest soils that found microbial P limitation rather than N limitation (e.g. Tischer et al., 2014; Wang et al., 2019; Jing et al., 2020). Across all geochemical regions, soil microbes remained P-limited at post-incubation (Fig. 4).

Even a possible release of organic P due to organic matter decomposition could not alleviate microbial P limitation in a 120-day incubation experiment. Since microbial P limitation in tropical soils results from long-term adaptation of soil microbial communities to site-specific soil and environmental conditions, EAA is not modified by short-term changes in nutrient availability (Jing et al., 2020). Thus, we argue that the alleviation of microbial P limitation would require a longer observation time than a 120-day incubation experiment with repeated inputs of organic matter. Despite markedly distinct patterns of total

P content (Doetterl et al., 2021) with highest values in the mafic region, patterns and magnitude of microbial P limitation did not differ strongly across geochemical regions. Within these undisturbed forest soils, processes regulating the P availability seem to be comparable between soils of contrasting geochemical origin, while processes regulating C availability are different.

**4.2 Subsoil microbes were less C-limited than their topsoil counterparts and inversely P-limited**

We found that subsoil microbes were less C-limited than their topsoil counterparts. In all three regions, vector length increasing

with C limitation was markedly highest in topsoils, particularly in the felsic and mafic regions (Fig. 4d−f). This is contrary to our expectation, that the highest C limitation would be found in subsoils due to low organic matter contents. High C availability in sub- compared to topsoils was evident with higher and sometimes similar $MBC_{DOC}$, $MBC_{SOC}$, and DOC:SOC (Fig. 1g−l, Table A2) in sub- compared to topsoils, particularly at pre-incubation. Even though DOC consists of a complex C pool (Kalbitz et al., 2003) that is not always easily available for microbial decomposition, our observation indicates that C solubility can be

high in subsoil, leading to potentially higher C availability (Wang et al., 2013; Salomé et al., 2010), and thus lower microbial C limitation. In tropical and subtropical forests of China, Jing et al. (2020) reported that microbes in subsoils are able to maintain their ability to mineralize soil organic matter and alleviate C limitation similarly to their topsoil counterparts. In our soils, Bukombe et al. (2021) observed similar or higher respiration ($CO_2$ $gC^{-1}$) in sub- compared to topsoils, particularly in the felsic region. This indicates that subsoil C was as accessible to microbes as was topsoil C under laboratory incubation

conditions. Although the mixed sedimentary region contained up to 52 % geogenic C, its respiration was highest in subsoils (Bukombe et al., 2021). Therefore, we assume that microbial communities in subsoils have developed strategies to live in low organic matter content environments and acquire C from less accessible C pools. Furthermore, conditions of high N availability as found in our study (Fig. 2,4) have been reported to increase microbial decomposition of tropical subsoil C (Meyer et al., 2018), and could thus also be a reason for the reduced C limitation. Nevertheless, the significantly lower $CO_2$ $gC^{-1}$ observed

by Bukombe et al. (2021) in sub- compared to topsoils of the mafic region seems to contradict our observation of low microbial C limitation. A possible explanation could be that low C respiration in subsoils of the mafic region is likely indicative of a high microbial carbon use efficiency as microbial strategy to reduce C loss in response to low organic matter content (Yuan et





al., 2019, Zhran et al., 2020). Overall, our findings suggest highest content of potentially labile C in subsoils, which could lead to markedly lowest C stock observed in subsoils of the felsic and mafic regions (Reichenbach et al., 2021). Hence, we postulate

that subsoils contain a relatively large SOC fraction with a potentially fast turnover which might limit the contribution of subsoils for organic C storage.

In contrast to lowest C limitation in subsoils, microbes were more P-limited than their topsoil counterparts (Fig. 5a–c). Most probably, recycling of organic matter is the major source of P in these deeply weathered tropical forest soils where subsoils are depleted in rock-derived nutrients (Chadwick and Asner, 2017). Hence, it is reasonable to find a high P limitation in subsoil

where organic matter content is low. Furthermore, relative to N and P, DOC was highest in subsoils (Table A2) indicating an increasing amount of potentially available C relative to N and P. The common understanding for younger (temperate zone) soils suggesting an increase in fresh minerals with soil depth that may favour the release of mineral P likely does not apply to these deeply weathered and undisturbed tropical forest soils. Overall, we conclude that unlike C, which can still be accessible to microbes in subsoils, P availability in deeply weathered tropical rainforest soils is much lower.

**4.3 Microbial biomass, carbon and phosphorus limitations were controlled by organic matter and soil depth**

Assessed physico-chemical soil properties significantly explained variance in MBC$_{Soil}$ at both pre- and post-incubation (Fig. 5). Over 45 % of explained MBC$_{Soil}$ variance was due to organic matter reflecting that resource abundance and availability are important factors driving the patterns of microbial biomass in tropical forest soils. The explanatory power of organic matter on MBC$_{Soil}$ was strongly influenced by soil depth. Mechanistically, organic matter content is tightly correlated to soil depth as

it declines with increasing soil depth and may interact differently with other soil geochemical properties and thus affect MBC$_{Soil}$. In contrast, MBC$_{SOC}$ was not driven by the assessed physico-chemical soil properties at pre-incubation, but was significantly controlled by soil acidity and exchangeable cations, solid phase geochemistry and substrate quality at post-incubation (Table 2). We propose that depletion in labile C over the course of incubation led to less abundant but more adapted microbial communities with wide stoichiometric requirements and able to live in reduced resource environments. This

microbial community could thus be the most affected by soil geochemistry. Whereas, the more abundant microbial biomass at pre-incubation was likely driven by higher moisture, temperature and oxygen conditions associated with a relatively higher amount of labile C. This possible shift in the microbial community is in line with less C limitation observed in topsoils at post-incubation, while the entire DOC pool declined compared to pre-incubation (Fig. 1d–f).

The relatively less predictable C and P limitations indicated that microbial resource acquisition is complex and could strongly

depend on microbial physiology and substrate stoichiometry, particularly at post-incubation (Stone et al., 2014; Cui et al., 2019). At pre-incubation, soil depth and organic matter were the most important explanatory variables for C limitation, explaining 48 % and 28 % of C limitation variance ($R^2 = 0.19$), respectively (Fig. 5, Table 2). The explanatory power of organic matter over C limitation disappeared after controlling for soil depth (Fig. 5). This observation indicates that changes in organic matter at depth could be both qualitative and quantitative and may lead to a shift in microbial resource acquisition strategy.

This is supported by higher C limitation observed in top- compared to subsoils, and higher P limitation observed inversely. At pre-incubation, organic matter was more important (37 % of contribution) than soil depth (29 %) in explaining variance in P



limitation at pre-incubation (Fig. 5). The strong explanatory power that organic matter has on P limitation supports our hypothesis that the major amount of P in these soils comes from organic matter cycling.

Overall, because tropical soils are deeply weathered, resource-related controls associated with organic matter abundance and
quality drove patterns of MBC$_{Soil}$, and at a lower extent C and P limitation, while soil geochemistry and substrate quality controlled MBC$_{SOC}$. Hence, control of soil parent material geochemistry on MBC$_{Soil}$ could be more indirect rather than direct, through its effects on resource availability, reflected here by MBC$_{SOC}$ (Insam and Domsch, 1988; Mendoza et al., 2020).

## 5 Conclusions and outlook

*Do patterns of microbial processes in deeply weathered tropical forest soils reflect the geochemical gradient of parent material*
*under similar climatic conditions?*

We conclude that patterns of microbial biomass (MBC$_{Soil}$) and enzyme activity in tropical soils of contrasting geochemical origin are markedly distinct due to differences in resource availability. In addition to the well-established microbial P limitation in tropical forest soils, we showed that microbial C and not P limitation differs between soil of contrasting geochemical origin. Similar patterns and magnitude of P limitation in these deeply weathered, undisturbed systems indicated that processes
regulating P availability do not differ between soils of contrasting geochemical origin.  Because of long-lasting efficient recycling of organic matter in humid tropics, none of the investigated regions and soil depths showed signs of N limitation for microbial processes.

*Are subsoil microbes in highly weathered tropical forest soils more C- and nutrient-limited than their topsoil counterparts?*

Less microbial C limitation in sub- compared to topsoils indicated that subsoil microbes can significantly participate in C
cycling and limit C storage if increased oxygen availability is prevalent. Furthermore, enhanced P limitation in sub- compared to topsoils indicated that the recycling of organic matter is the major P source.

The common understanding for younger (temperate zone) soils suggesting an increase in fresh minerals at increasing soil depth that may favour a release of mineral P does likely not apply to these deeply weathered tropical forest soils where also subsoils are depleted in rock-derived nutrients.

*What are important controls on soil microbial biomass, C and nutrient limitations?*

We conclude that microbial biomass C (MBC$_{Soil}$) is predominantly driven by organic matter which constitutes the main resource for soil microbes. Control of organic matter over MBC$_{Soil}$ is strongly influenced by soil depth which could define the quality and quantity of organic matter as well as its interaction with other geochemical soil properties. In contrast, soil geochemistry and substrate quality are significant controls for MBC$_{SOC}$ particularly under conditions of less labile C. Microbial
C and P limitations depend on other controls than those assessed in this study. Although relatively weak, explained variance in C and P limitations was mainly related to organic matter and its dependence on soil depth.



## Outlook

Our findings point to the need to consider the variability in soil parent material in tropical pristine forests when assessing soil microbial processes in organic matter turnover models. Insights about possible shifts in the microbial communities across geochemical regions and soil depth are needed to explain changes in microbial nutrient limitation.

**Data availability.** The lab data used in this study are part of a database publication by Doetterl et al. (2021, same special issue), it is also available from the corresponding author upon reasonable request.

**Author contribution.** CV, KK and SD designed the research. SD acquired funding for the research. LKK and SD conducted the sampling campaign. LKK, FKO and CV conducted lab experiments and analyses. LKK, CV, KK and SD analysed and interpreted the data. All authors contributed to the writing of the paper under the lead of LKK.

**Competing interests.** The authors declare that they have no conflict of interest.

**Acknowledgments.** The authors thank following collaborators of the project TROPSOC: International Institute of Tropical Agriculture (IITA) in DR Congo, Institute of Geography at Augsburg University, Faculty of Agricultural Sciences at the Catholic University of Bukavu, School of Agricultural and Environmental sciences at the Mountains of the Moon University, Institut Congolais pour la Conservation de la Nature (ICCN), and Rwanda Development Board (RDB). The authors also thank the whole TROPSOC team, student assistants, and lab technicians for their important work in the laboratory and all fieldwork helpers who made the sampling campaign possible. Special thanks go to Gina Garland for language editing. Finally, authors thank the German Academic Exchange Service (DAAD) for financing the research stay of LKK in Germany.

**Financial support.** This work is part of the DFG funded Emmy Noether Junior Research Group "Tropical soil organic carbon dynamics along erosional disturbance gradients in relation to soil geochemistry and land use" (TROPSOC; project number 387472333).

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

## 12 Appendix

**Table A1.** Effect of topography on investigated microbial variables. Comparison is made using a two-way ANOVA by comparing topographic positions (i.e. plateau, slopes, and valleys) separately for each geochemical region.

| Variables | Pre-incubation | | | Post-incubation | | |
|---|---|---|---|---|---|---|
| | Mafic | Mixed sedimentary | Felsic | Mafic | Mixed sedimentary | Felsic |
| | p-value | | | | | |
| C acquiring enzymes | 0.89 | 0.35 | 0.41 | 0.29 | 0.59 | 0.75 |
| N acquiring enzymes | 0.04 | 0.09 | 0.13 | 0.74 | 0.16 | 0.12 |
| P acquiring enzymes | 0.91 | 0.88 | 0.36 | 0.79 | 0.65 | 0.14 |
| Vector length | 0.85 | 0.07 | 0.86 | 0.66 | 0.71 | 0.40 |
| Vector angle | 0.03 | 0.02 | 0.05 | 0.85 | 0.31 | 0.39 |
| Microbial biomass C | 0.33 | 0.50 | 0.07 | 0.21 | 0.68 | 0.10 |





**Table A2.** Means and standard errors of dissolved organic C (DOC) related to soil organic C (DOC:SOC), total dissolved N (DOC:TDN) and bioavailable P (DOC:BP) across geochemical regions and soil depths at pre-incubation.

| Geochemical regions | Soil depths [cm] | DOC:SOC [mg g$^{-1}$] | DOC:TDN [mg mg$^{-1}$] | DOC:BP [mg mg$^{-1}$] |
|---|---|---|---|---|
| | 0-10 | 5.5 ± 0.7 | 1.5 ± 0.2 | 11.3 ± 1.1 |
| Mafic | 30-40 | 6.0 ± 0.6 | 3.5 ± 0.3 | 15.6 ± 3.0 |
| | 60-70 | 5.8 ± 1.0 | 4.4 ± 0.5 | 14.3 ± 5.5 |
| | 0-10 | 5.8 ± 0.7 | 2.6 ± 0.3 | 80.1 ± 21.7 |
| Mixed sedimentary | 30-40 | 3.9 ± 0.4 | 5.7 ± 0.9 | 77.4 ± 31.5 |
| | 60-70 | 3.9 ± 1.0 | 8.4 ± 1.8 | 116.1 ± 53.6 |
| | 0-10 | 3.5 ± 0.3 | 0.9 ± 0.1 | 8.4 ± 1.5 |
| Felsic | 30-40 | 4.0 ± 0.5 | 3.2 ± 0.5 | 12.8 ± 4.8 |
| | 60-70 | 5.2 ± 0.9 | 6.3 ± 0.8 | 9.9 ± 3.8 |