# Peer review of "Patterns of microbial processes shaped by parent material and soil depth in tropical rainforest soils"

_SOIL, 2020_

## Referee Comment (RC1) · Anonymous Referee #1 · 6 Dec 2020

This is a conceptually straightforward incubation study seeking insight to the effects of soil parent material geochemistry on soil microbial biomass and extracellular enzyme activity in tropical Africa, where limited research has been conducted previously. The general results were that SOM and depth were the most important explanatory variables for MBC (and C-limitation), soils were consistently P-limited and P-limitations were strongly related to SOM, and not surprisingly, SOM was related to depth. The conclusion was that soil geochemistry affected MBC indirectly through affecting resource availability. Although analyses were detailed and meticulous, the results provided little novel insight and are generally consistent with other studies of resource limitations in tropical soils. Thus, microbial systems in tropical soils in Africa appear to have similar

constraints as elsewhere. The paper provides a solid background for this system, but the discussion could be much shorter with greater emphasis on similarities to other tropical systems.

Section 2.5. The calculations of vector characteristics of extracellular enzyme activities were based on log ratios rather than proportions suggested by Moorhead et al. (2016). This choice affects results and warrants a brief explanation of choice. Moreover, both equations 1 and 2 seem to have mathematical errors. Equation 1 lists ln(BG+CB)/ln(LAP)^2, that should be (ln(BG+CB)/ln(LAP))^2, whereas equation 2 lists ln(BG+CB/ln(LAP), that should be ln(BG+CB)/ln(LAP), and ln(BG+CB/ln(NAG+LAP)), which should be ln(BG+CB)/ln(NAG+LAP). Please confirm and correct.

Section 3.3. There is no absolute vector threshold for C, N or P limitation, only relative indications within a study. For example, vector (relative EEA) characteristics do not directly reflect availability of C, N or P that does not require enzyme action to acquire, only indirect evidence of such availability through relatively lower enzyme activity needed to acquire it from hydrolysable sources.

The convergence in post incubation vector length patterns with depth across the three soils indicates a similar balance of enzyme-driven C, N and P acquisition despite differences in other soil characteristics. This is interesting in light of the substantial geochemical differences between soils.

Section 3.5. The correspondences between predictions and observations are not convincing.

Section 4.1. It was hard to follow this discussion, but the general relationship between MBC and resource availability (C and N) seems to contradict the following paragraph stating that microbes were P-limited throughout. Also, the implication that EEA is not responsive to short term variations in resource availability questions those parts of this study.

Section 4.3. This paragraph reiterated the relationship between MBC and SOM, mentioned earlier, and responsive to soil characteristics that vary with site and depth. This is consistent with many other studies. The most important point arrives at lines 389-391, i.e., relative resource limitation is complicated.

---

## Referee Comment (RC2) · Anonymous Referee #2 · 14 Feb 2021

This manuscript explores how soil geochemistry (parent material) influences microbial functions in weathered tropical rainforest. To do so, the authors used the classical soil forming factors as an approach. They assume that time for soil development, climate, topography and biota are kept constant, and only parent material (geology) varies (mafic, mixed and felsic). However, as a reader I miss information on biota. Vegetation is "dense tropical forest" (line 91, Wilken 2021, missing in the references), but is soil fauna actually identical? Does the microbial community composition change? Do plant species differ? The authors should clarify why they believe that the factor "biota" actually is kept constant across the catena and different parent material. After all, the whole methodological approach is based on the assumption that all soil forming

factors are kept identical, except parent material.

Overall, I found it difficult to identify a key message of the manuscript. Maybe it would be good to simplify and shorten (39%) the text, show less figures, and focus the conclusions on the results on the results shown in the manuscript.

There is some potential for clarifications:

Introdution – briefly explain the differences between mafic and felsic

Sample handling – the soil samples were air-dried (line 117), which typically should be avoided for reliable analysis soil microbial biomass and extracellular enzyme activity.

Statistics and data presentation – The data presentation is difficult to follow, e.g. the choice of the vector analysis warrants an explanation. How can the p-value be 0.00 (line 277; Fig 5)?.

Results - The results are consistent with general knowledge that tropical soils are P-limited. The authors should emphasis on similarities/differences to other tropical systems.

Discussion - Typically manuscripts point out potential methodological limitations, and discuss how these limitations affect the outcome. The addition of such a section would strengthen the credibility of the discussed implications.

Literature citations – some of the citations used are not include in the reference. Unpublished works are referenced multiple times but it is difficult to assess the statements in the manuscripts based on unpublished works.

Conclusion section – much of this is a stretch from the data. The conclusion section should be refocused on the data that is presented within the manuscript.

Some more thoughts and questions: How much C was respired as $CO_2$ over the incubation across treatments? (LINE 125) Did this vary statistically across treatments, and if so, how? Line 96. Spelling of schist Line 190. Spelling of bestNormalize Line 190.

Which transformation was perfomed on which variables? Line 216. Define MBCDOC Line 314. EAA or EEA? Fig 1. The figures should be set up in more comprehensive way, with abbreviations defined and treatments given in the same order and with legends. Fig 5. Are these p values significantly different?

---

## Author Comment (AC1) · 28 Mar 2021

**Point-by-point responses to Referee's comments**

The Authors thank the Anonymous Referee #2 for the thorough review and valuable constructive comments. We addressed the comments and provided suggestions for the revision of the manuscript. New text to be added/modified in the manuscript is blue-coloured in this letter.

**Referee comment 1**. This is a conceptually straightforward incubation study seeking insight to the effects of soil parent material geochemistry on soil microbial biomass and extracellular enzyme activity in tropical Africa, where limited research has been conducted previously. The general results were that SOM and depth were the most important explanatory variables for MBC (and C-limitation), soils were consistently P-limited and P-limitations were strongly related to SOM, and not surprisingly, SOM was related to depth. The conclusion was that soil geochemistry affected MBC indirectly through affecting resource availability.

Although analyses were detailed and meticulous, the results provided little novel insight and are generally consistent with other studies of resource limitations in tropical soils. Thus, microbial systems in tropical soils in Africa appear to have similar constraints as elsewhere.

*Authors' response 1. Thank you for this comment. We agree that microbial processes in tropical soils of Africa have similar constraints as elsewhere (e.g., P but not N limitation). However, we showed that the geochemistry of the parent material shaped vertical patterns of C and P limitations in tropical systems. Overall, despite the long-lasting chemical weathering of investigated soils, the geochemistry of the parent material still affects resource availability which in turn shape patterns of $MBC_{Soil}$ and EEA. This is an element that has so far not been highlighted enough in research on tropical biogeochemistry. Following changes will be made in the discussion section:*

*In contrast to the lowest C limitation in subsoils, microbes were more P-limited than their topsoil counterparts (Fig. 3a–c). Most probably, recycling of organic matter is the major source of P in these deeply weathered tropical forest soils where subsoils are depleted in rock-derived nutrients (Chadwick and Asner, 2017). Hence, it is reasonable to find a high P limitation in subsoil where organic matter content is low. Furthermore, relative to N and P, DOC was highest in subsoils (Table A2) indicating an increasing amount of potentially available C relative to nutrients. Comparable depth gradient of P limitation was reported by Jing et al. (2020) in tropical forests of China. However, Lui et al. (2020) showed similar or lower P limitation in top- compared to subsoils in subtropical forests of China. Overall, while this study confirms the well-established P limitation rather than N in the humid Tropics, it reveals different vertical patterns of C and P limitations compared to other tropical regions likely due to differences in the geochemistry of the parent material. Moreover, the common understanding for younger (temperate zone) soils suggesting an increase in fresh minerals with soil depth that may favour the release of mineral P likely does not apply to these deeply weathered tropical forest soils.*

**Referee's comment 2.** The paper provides a solid background for this system, but the discussion could be much shorter with greater emphasis on similarities to other tropical systems.

*Authors' response 2. We will shorten the discussion as much as possible without constraining the interpretation and explanation of our results too much. Specifically, section 4.3. will be discussed together with section 4.1 as both sections discuss relationships between resource availability and microbial processes. To shorten the text as also suggested by Referee #2, we will move Fig. 2 to the supplement as it is cited only once in the discussion and remove Fig. 5 as its information could be contained in Table 2. We will emphasize both similarities and differences of our findings to other tropical regions. We will focus our comparisons on studies that assessed microbial C and nutrient limitations using vector analysis. Comparison of our findings with other studies will be done as follow:*

*Physico-chemical soil properties measured in this study did not greatly explain variance in C and P limitations (Table 2 suggesting that microbial resource acquisition is complex and could strongly*

*depend on microbial physiology and substrate stoichiometry (Stone et al., 2014; Cui et al., 2019). In subtropical forests of China, Lui et al. (2020) revealed that soil C, N, and P stoichiometry was an important factor defining microbial C and nutrient limitations. Phosphorous rather than N limitation as observed in this study is usually attributed to weathering losses of P-containing primary minerals and P occlusion by secondary minerals in tropical soils (Chadwick et al., 1999). Also, due to the long-lasting efficient recycling of organic matter, tropical forest soils are rarely N-limited (Chadwick and Asner, 2017). Although this finding is not new; it corroborates several studies in tropical and subtropical forest soils that found microbial P limitation rather than N limitation (e.g., Camenzind et al., 2018; Wang et al., 2019; Jing et al., 2020). Across all geochemical regions, soil microbes remained P-limited at post-incubation (Fig. 3). Even a possible release of P due to organic matter decomposition could not alleviate microbial P limitation in a 120-day incubation experiment. According to Jing et al. (2020), microbial P limitation in tropical soils results from long-term adaptation of soil microbial communities to site-specific soil and environmental conditions. We argue that the alleviation of microbial P limitation would require a longer observation time than a 120-day incubation experiment with repeated inputs of organic matter. In the study by Wang et al. (2020) a 10-year P addition experiment was necessary to reduce microbial investments towards enzyme production and thus alleviated microbial P limitation in tropical forest soils.*

**Referee's comment 3.** Section 2.5. The calculations of vector characteristics of extracellular enzyme activities were based on log ratios rather than proportions suggested by Moorhead et al. (2016). This choice affects results and warrants a brief explanation of choice.

***Authors' response 3.*** *Thank you for this comment which was very helpful to improve our manuscript. We recalculated vector characteristics using untransformed proportions. This method yielded lower skewness and kurtosis than log-transformed ratios (Table 1).*

**Table 1.** Skewness and kurtosis of untransformed and log-transformed vector characteristics

|  | Vector length | | Vector angle | |
| --- | --- | --- | --- | --- |
| Untransformed proportions | Pre-incubation | Post-incubation | Pre-incubation | Post-incubation |
| Skewness | -0.95 | -0.30 | -0.36 | -1.12 |
| Kurtosis | 0.61 | -0.76 | -0.65 | 1.15 |
| Log-transformed ratios | | | | |
| Skewness | 5.24 | 6.02 | 1.79 | 2.14 |
| Kurtosis | 31.13 | 38.72 | 3.95 | 6.27 |

*Moreover, untransformed proportional ratios of enzyme activity allowed to eliminate undefined values resulting from a zero in the denominator or in a logarithmic transformation (Moorhead et al. 2016). Therefore, we will use untransformed proportions in this study. Even though the obtained vector values differ between these two methods, the trend and main conclusions remain the same (Fig. 1).*

[Figure]

**Figure 1.** Vector analysis of microbial C and nutrient limitations. Error bars are standard errors of means (n=12). Vector length increases with microbial C limitation. Vector angle with <45° indicates N and >45° indicates P limitation. Lower-case letters compare geochemical regions per soil depth at pre-incubation. Upper-case letters and asterisks (*) compare geochemical regions per soil depth at post-incubation.

**Referee's comment 4.** Moreover, both equations 1 and 2 seem to have mathematical errors. Equation 1 lists ln(BG+CB)/ln(LAP)^2, that should be (ln(BG+CB)/ln(LAP))^2, whereas equation 2 lists ln(BG+CB/ln(LAP), that should be ln(BG+CB)/ln(LAP), and ln(BG+CB/ln(NAG+LAP)), which should be ln(BG+CB)/ln(NAG+LAP). Please confirm and correct.

*Authors' response 4. Thank you for this hint. Since we will change the calculation of vector characteristics by using untransformed proportional ratios, the formula will be changed in the manuscript accordingly. The new formula will be:*

$$Vector\ length = \sqrt{X^2 + Y^2} \tag{1}$$

$$Vector\ angle\ (°) = Degrees(Atan2(X,Y)) \tag{2}$$

*Where,*

$$X = \frac{CB+BG}{(CB+BG)+(AP)}, \ Y = \frac{CB+BG}{(CB+BG)+(NAG+LAP)}$$

*Note. CB: Cellobiohydrolase, BG: β-glucosidase, NAG: N-acetylglucosaminidase, LAP: leucine-aminopeptidase*

**Referee's comment 5.** Section 3.3. There is no absolute vector threshold for C, N or P limitation, only relative indications within a study. For example, vector (relative EEA) characteristics do not directly reflect availability of C, N or P that does not require enzyme action to acquire, only indirect evidence of such availability through relatively lower enzyme activity needed to acquire it from hydrolysable sources.

*Authors' response 5. We agree that vector characteristics reflect indirectly CNP availability and thus relative microbial C and nutrient limitations. We will clearly mention it throughout the manuscript as follow:*

*"Exploiting the enzyme data in a comprehensive and cost-effective way for assessing nutrient limitations we used vector characteristics (e.g., vector length and vector angle) to evaluate relative microbial C and nutrient limitations (Moorhead et al., 2013; Hill et al., 2014; Moorhead et al., 2016)".*

*"Vector analysis reflects relative microbial investments in C, N, and P acquisition (Moorhead et al., 2013). Vector length increased with relative microbial C vs. nutrient limitation. Vector angle indicated relative P vs. N limitation, with angle <45° indicating N and angle >45° indicating P limitation (Moorhead et al., 2013; Chen et al., 2018; Cui et al., 2019)".*

*"Across all geochemical regions, the vector angle was >45° at all soil depths and indicated relative microbial P limitation rather than N limitation (Fig. 3 a–c). At pre-incubation, vector angle in topsoils was lowest in the mafic region and highest in mixed sedimentary region. The vector angle was higher in sub- (30-40cm) than topsoils at pre-incubation for all geochemical regions (p<0.05) but similar among soil depth and geochemical regions at post-incubation. At pre-incubation, vector length indicating relative microbial C limitation was highest in the mixed sedimentary region and lowest in the felsic region for both top- and subsoils (p<0.05). At post-incubation, microbial C limitation declined in topsoils and was similar among geochemical regions and soil depths (p>0.05)".*

**Referee's comment 6.** The convergence in post incubation vector length patterns with depth across the three soils indicates a similar balance of enzyme-driven C, N and P acquisition despite differences in other soil characteristics. This is interesting in light of the substantial geo- chemical differences between soils.

*Authors' response 6. Thank you for this highlight. We will add the following sentences to mention it in the manuscript.*

*Despite differences in soil geochemistry and resource availability, microbial C and P limitations were similar among geochemical regions and soil depths at post-incubation. This observation suggests a similar balance of enzyme-driven C, N and P acquisition in these soils likely caused by substrate depletion during incubation.*

**Referee's comment 7.** Section 3.5. The correspondences between predictions and observations are not convincing.

*Authors' response 7. Thank you for this comment. We would like to point out that correspondences between predictions and observations showing a quite low prediction power of our independent variables are a result of high variability in the dataset (i.e., differing strength of correlations between independent and dependent variables across geochemical regions). In the mixed sedimentary region, for instance, C was poorly correlated to absolute microbial biomass due to the high content of geogenic organic C which is poorly accessible to microbes (Reichenbach et al. 2021, in review) while strong correlations were observed in the felsic and mafic regions. Therefore, observed correspondences between predicted and observed MBC resulted in relatively low $R^2$. Concerning predictions of relative C and P limitations, our data are an indication that microbial resource acquisition is complex and could strongly depend on other factors such as microbial physiology and substrate stoichiometry.*

**Referee's comment 8.** Section 4.1. It was hard to follow this discussion, but the general relationship between MBC and resource availability (C and N) seems to contradict the following paragraph stating that microbes were P-limited throughout. Also, the implication that EEA is not responsive to short term variations in resource availability questions those parts of this study.

*Authors' response 8. Thank you for this comment. We will re-write this part of the discussion in order to address comment 2 and 9. Concerning the relationship between microbial biomass C ($MBC_{Soil}$) and resource availability, we show that $MBC_{Soil}$ follows not only patterns in C and N abundance but also C quality. Higher C content was not always associated with higher $MBC_{Soil}$. This was the case in the mixed sedimentary region where poorly available geogenic organic C was found. Microbial biomass differed between studied soils due to differences in C and N resource quantity and quality. Concerning P limitation, the fact that microbes are P-limited elsewhere is a general observation for studied soils. However, the magnitude of P limitation still differs between compared soils particularly at pre-incubation in topsoils. This observation is in line with the observed relationship between $MBC_{Soil}$ and resource availability. Please see below the proposed revision of the first section of the discussion:*

*Investigated geochemical regions showed contrasting C and N contents that defined patterns of MBC$_{Soil}$ and EEA with the highest values in the mafic region (Fig. 1-2, Fig. A1). This is in line with our multiple regression analysis indicating that organic matter is the main explanatory variable driving MBC$_{Soil}$ at both pre- and post-incubation (Table 2). Observed patterns in C and N contents among geochemical regions likely reflect differences in organic matter input and its interactions with geochemical soil properties regulating its stabilization. Highest C stabilization potential, favoured by highest pedogenic oxides was reported as a strong driver of C stock in the mafic region (Reichenbach et al., 2021). Conversely, low pedogenic oxide contents in the mixed sedimentary region limit C stabilization potential and C stock (Reichenbach et al. (2021), leading to lower MBC$_{Soil}$ and EEA. In the mixed sedimentary region, poor C quality limited MBC$_{Soil}$ independently of C quantity. The mixed sedimentary region showed over 2 times more DOC than the felsic region, however over 0.5 times less MBC$_{Soil}$ in topsoils (Fig. 1 a–f). This observation is due to the presence of geogenic organic C (up to 52 %) which is poorly available to microbes due to its chemical composition (Reichenbach et al., 2021). Low C availability in the mixed sedimentary region is supported by the observed lowest MBC$_{DOC}$ (Fig. 1). Indeed, small amounts of MBC within a large pool of C mean that the average availability of C sources must be low (Insam and Domsch, 1988). In contrast MBC$_{Soil}$, MBC$_{SOC}$ was highest in the felsic region which indicates high C availability in these soils and supports a lack of strong C stabilization mechanisms. The most important controls on MBC$_{SOC}$ were soil acidity and exchangeable cations, solid phase geochemistry and substrate quality at post-incubation but not at pre-incubation (Table 2). We propose that depletion in labile C over the course of incubation led to less abundant but more adapted microbial communities with wide stoichiometric requirements and able to live in environments with reduced resources. This microbial community established in environments with little available resources could thus be the most affected by soil geochemistry. Whereas the more abundant microbial biomass at pre-incubation was likely driven by higher moisture, temperature and oxygen conditions associated with a relatively higher amount of labile C. Overall, despite the long-lasting chemical weathering of investigated soils, the geochemistry of the parent material still affects resource availability which in turn shape patterns of MBC$_{Soil}$ and EEA.*

*Soil microbes were P-limited rather than N-limited throughout all three regions (Fig. 3). The lowest P limitation was observed in the mafic region particularly in topsoils. This is in line with the highest P content, which is due to higher organic matter content (Doetterl et al., 2021). However, the similar magnitude of P limitation among geochemical regions in subsoils indicates that within these undisturbed forest subsoils, processes regulating the P availability seem to be comparable between soils of contrasting geochemical origin.*

**Referee's comment 9.** Section 4.3. This paragraph reiterated the relationship between MBC and SOM, mentioned earlier, and responsive to soil characteristics that vary with site and depth. This is consistent with many other studies. The most important point arrives at lines 389- 391, i.e., relative resource limitation is complicated.

*Authors' response 9. To address this comment, we will merge sections 4.1. and 4.3. as both sections discuss relationships between resource availability and microbial processes.*

**References**

Moorhead, D.L., Rinkes, Z.L., Sinsabaugh, R.L., Weintraub, M.N.: Dynamic relationships between microbial biomass, respiration, inorganic nutrients and enzyme activities: informing enzyme-based decomposition models, Front. Microbiol., 4, 223, https://doi.org/10.3389/fmicb.2013.00223, 2013.

Moorhead, D. L., Sinsabaugh, R. L., Hill, B. H., Weintrau, M. N.: Vector analysis of ecoenzyme activities reveal constraints on coupled C, N and P dynamics, Soil Biol. Biochem., 93, 1-7, http://dx.doi.org/10.1016/j.soilbio.2015.10.019, 2016.

Jing, X., Chen, X., Fang, J., Ji, C., Shen, H., Zheng, C., Zhu, B.: Soil microbial carbon and nutrient constraints are driven more by climate and soil physicochemical properties than by nutrient addition in forest ecosystems, Soil Biol. Biochem., 141, 107657. https://doi.org/10.1016/j.soilbio.2019.107657, 2020.

Liu, J., Chen, J., Chen, G., Guo, J., Li, Y.: Enzyme stoichiometry indicates the variation of microbial nutrient requirements at different soil depths in subtropical forests, PLoS ONE 15, e0220599, https://doi.org/10.1371/journal.pone.0220599, 2020.

Wang, S., Mori, T., Mo, J., Zhang, W.: The responses of carbon- and nitrogen-acquiring enzymes to nitrogen and phosphorus additions in two plantations in southern China, J. Forest Res., 31, 319–1324 https://doi.org/10.1007/s11676-019-00905-0, 2019.

---

## Author Comment (AC2) · 28 Mar 2021

**Point-by-point responses to Referee's comments**

The Authors thank the Anonymous Referee #2 for the thorough review and valuable constructive comments. We addressed the comments and provided suggestions for the revision of the manuscript. New text to be added/modified in the manuscript is blue-coloured in this letter.

**Referee comment 1.** This manuscript explores how soil geochemistry (parent material) influences microbial functions in weathered tropical rainforest. To do so, the authors used the classical soil forming factors as an approach. They assume that time for soil development, climate, topography and biota are kept constant, and only parent material (geology) varies (mafic, mixed and felsic). However, as a reader I miss information on biota. Vegetation is "dense tropical forest" (line 91, Wilken 2021, missing in the references), but is soil fauna actually identical? Does the microbial community composition change? Do plant species differ? The authors should clarify why they believe that the factor "biota" actually is kept constant across the catena and different parent material. After all, the whole methodological approach is based on the assumption that all soil forming factors are kept identical, except parent material.

*Authors' response 1. Thank you for pointing this out. We did not analyse the microbial community composition. Our choice for the study sites has been motivated by a prior knowledge of vegetation structure, climate, and topography that are comparable. Overall, soil forming factors are similar among our sites except for the parent material. As different parent materials lead to different soil development which in turn affect soil biota, we can expect that microbial community differ as they have to adapt to soil conditions such as pH, nutrient availability, and texture (Fierer and Jackson, 2006; Deng et al., 2015; Fierer, 2017). Therefore, we assume that parent material geochemistry would shape the patterns of microbial processes in studied sites.*

**Referee comment 2.** Overall, I found it difficult to identify a key message of the manuscript. Maybe it would be good to simplify and shorten (39%) the text, show less figures, and focus the conclusions on the results on the results shown in the manuscript.

*Authors' response 2. Thank for this suggestion. We will shorten the manuscript (mainly the result, discussion, and conclusion sections) as much as possible without constraining the interpretation and explanation of our results too much. To shorten the text, we will move Fig. 2 to the supplement as it was cited only once in the discussion. Additionally, we will remove Fig. 5 as its information could be contained in Table 2. As also suggested by Referee #1, we will merge sections 4.1. and 4.3. as both sections discuss relationships between resource availability and microbial processes.*

*Changes we will make mainly in the discussion and conclusion sections (following comments from the Referee #1 and #2) will allow us to improve the clarity of the key message. "Briefly, despite the advanced weathering stage of tropical rainforest soils, variability of the parent material still affects microbial processes through its effect on resource availability. This is an element that has so far not been highlighted enough in research on tropical biogeochemistry. Specifically, we emphasize on patterns of microbial resource limitations across contrasting geochemical regions and soil depths for understanding organic matter decomposition in tropical Africa where such studies are scarce. Furthermore, the manuscript reveals the weak effect of geochemical soil properties in controlling microbial C and P limitations indicating that microbial resource acquisition is complex".*

**Referee comment 3.** There is some potential for clarifications

Introduction – briefly explain the differences between mafic and felsic.

*Authors' response 3. Thank for this suggestion. In addition to bedrock and soil samples description provided in the method section (Section 2.1), we will add the following sentences in the introduction:*

*We used soil materials from three geochemically distinct parent materials (from mafic to felsic) and three soil depths (0−70 cm) and measured microbial biomass C and enzyme activity (i.e. C, N, and P acquiring enzymes) at the beginning and end of a 120-day incubation experiment. Mafic describes igneous rocks composed of minerals that are rich in iron and magnesium, dense, and typically dark in colour. It is contrasted with felsic which describes igneous rocks composed of minerals such as quartz and feldspar and are relatively light in colour and density (Panchuk, 2019).*

**Referee comment 4.** Sample handling – the soil samples were air-dried (line 117), which typically should be avoided for reliable analysis soil microbial biomass and extracellular enzyme activity.

***Authors' response 4.*** *Thank you for this comment. We agree that fresh soil samples are preferred for microbial analyses. However, logistic and working conditions in tropical Africa could not allow us to keep the samples under their field conditions (e.g., by freezing them or conducting analyses immediately after the sampling) as our analyses had to be conducted in Europe. As indicated in the manuscript (section 2.3), microbial activities were measured on re-wetted soils (i.e., both pre-incubated and incubated samples).*

**Referee comment 5.** Statistics and data presentation – The data presentation is difficult to follow, e.g., the choice of the vector analysis warrants an explanation. How can the p-value be 0.00 (line 277; Fig 5)?

***Authors' response 5.*** *We will provide the explanation for the choice of vector analysis in the manuscript as follow:*

*Exploiting the enzyme data in a comprehensive and cost-effective way for assessing nutrient limitations we used vector characteristics to evaluate relative microbial C and nutrient limitations (Moorhead et al., 2013; Hill et al., 2014).*

*Concerning p-value, we will correct the sentence as follow: Soil depth together with physico-chemical properties explained significantly ($p<0.05$) the variance in $MBC_{Soil}$ and $MBC_{SOC}$ at pre- and post-incubation and only at pre-incubation for C and P limitations (Table 2).*

**Referee comment 6.** Results - The results are consistent with general knowledge that tropical soils are P- limited. The authors should emphasis on similarities/differences to other tropical systems.

***Authors' response 6.*** *We will emphasize on both similarities and differences of our findings to other tropical regions. We will focus our comparisons on studies that assessed microbial C and nutrient limitations using vector analysis. These changes will be made as follow:*

*In contrast to the lowest C limitation in subsoils, microbes were more P-limited than their topsoil counterparts (Fig. 3a–c). Most probably, recycling of organic matter is the major source of P in these deeply weathered tropical forest soils where subsoils are depleted in rock-derived nutrients (Chadwick and Asner, 2017). Hence, it is reasonable to find a high P limitation in subsoil where organic matter content is low. Furthermore, relative to N and P, DOC was highest in subsoils (Table A2) indicating an increasing amount of potentially available C relative to nutrients. Comparable depth gradient of P limitation was reported by Jing et al. (2020) in tropical forests of China. However, Lui et al. (2020) showed similar or lower P limitation in top- compared to subsoils in subtropical forests of China. Overall, while this study confirms the well-established P limitation rather than N in the humid Tropics, it reveals different vertical patterns of C and P limitations compared to other tropical regions likely due to differences in the geochemistry of the parent material. Moreover, the common understanding for younger (temperate zone) soils suggesting an increase in fresh minerals with soil depth that may favour the release of mineral P likely does not apply to these deeply weathered tropical forest soils.*

*Physico-chemical soil properties measured in this study did not greatly explain variance in C and P limitations (Table 2 suggesting that microbial resource acquisition is complex and could strongly depend on microbial physiology and substrate stoichiometry (Stone et al., 2014; Cui et al., 2019). In*

*subtropical forests of China, Lui et al. (2020) revealed that soil C, N, and P stoichiometry was an important factor defining microbial C and nutrient limitations. Phosphorous rather than N limitation as observed in this study is usually attributed to weathering losses of P-containing primary minerals and P occlusion by secondary minerals in tropical soils (Chadwick et al., 1999). Also, due to long-lasting efficient recycling of organic matter, tropical forest soils are rarely N-limited (Chadwick and Asner, 2017). Although this finding is not new; it corroborates several studies in tropical and subtropical forest soils that found microbial P limitation rather than N limitation (e.g., Camenzind et al., 2018; Wang et al., 2019; Jing et al., 2020). Across all geochemical regions, soil microbes remained P-limited at post-incubation (Fig. 3). Even a possible release of P due to organic matter decomposition could not alleviate microbial P limitation in a 120-day incubation experiment. According to Jing et al. (2020), microbial P limitation in tropical soils results from long-term adaptation of soil microbial communities to site-specific soil and environmental conditions. We argue that the alleviation of microbial P limitation would require a longer observation time than a 120-day incubation experiment with repeated inputs of organic matter. In the study by Wang et al. (2020) a 10-year P addition experiment was necessary to reduce microbial investments towards enzyme production and thus alleviated microbial P limitation in tropical forest soils.*

**Referee comment 7.** Discussion - Typically manuscripts point out potential methodological limitations, and discuss how these limitations affect the outcome. The addition of such a section would strengthen the credibility of the discussed implications.

***Authors' response 7.*** *Thank you for this hint. Since our discussion is already quite long, we will add the following section as a supplement and refer to it in the method section.*

***Limits of vector analysis in assessing microbial resource limitations***

*Our method for assessing microbial C and nutrient limitations (i.e., vector analysis) is based on stoichiometric and metabolic theories of ecological systems (Allison et al., 2011). Here, EEA is assumed to reflect microbial resource demand as enzyme production is regulated by microbial stoichiometric and metabolic requirements for resources (i.e., C, N, and P) and their environmental availability. However, a substantial fraction of soil enzymes could be constitutively produced (or stabilized on mineral surfaces) and thus obscures short-term shifts in microbial enzyme allocation in response to changes in nutrient availability (Allison and Vitousek, 2005; Geisseler et al., 2010). In this study, we observed that despite the increase in N content (Fig. A1), N acquiring enzyme activity remained high (Fig. 2), which likely affected the magnitude of P limitation relative to N as observed at post-incubation (Fig. 3). This observation implies that when significant amounts of enzymes can be out of prompt microbial regulation, the interpretation of microbial resource limitation may require caution, which could be the case for our post-incubation results. The vector analysis reveals C limitation relative to N and P and P vs. N limitations rather than absolute limitations of these nutrients. Furthermore, readily available C, N, and P that can be assimilated without enzyme actions cannot be directly reflected by vector angle and length but only indirectly by lower EEA. The vector analysis is useful for understanding microbial resource acquisition and organic matter decomposition although it provides only indirect evidences of environmental resource availability and relative microbial resource limitations (Hill et al., 2014). Another limit of the method is that N acquiring enzymes (e.g., NAG and LAP) can also generate C that is not reflected in the calculation of vector analysis and thus affect C/N enzymatic ratio differently from C/P enzymatic ratio (Moorhead et al., 2016). This issue is usually reported in studies that measured only BG as C acquiring enzyme (e.g., Moorhead et al., 2016; Wang et al., 2018; Mori et al., 2018). Including CB together with BG in the calculation of vector characteristics as done in this study could reduce the error arising from C acquired by N acquiring enzymes.*

**Referee comment 8.** Literature citations – some of the citations used are not include in the reference. Unpublished works are referenced multiple times but it is difficult to assess the statements in the manuscripts based on unpublished works.

*Authors' response 8. Thanks a lot for this hint. We apologize for this and we will carefully go through the manuscript and include all missing citations in the reference list. The missing references are related studies to be published in the same special issues with our study that got no DOI yet. All of these studies are already posted as preprints in SOIL discussion (e.g., Bukombe et al., 2021; Reichenbach et al., 2021; Wilken et al., 2021) and GFZ Data Services (Doetterl et al., 2021) and got their DOIs in the meanwhile. Please see references below:*

Doetterl, S.; Bukombe, B.; Cooper, M.; Kidinda, L.; Muhindo, D.; Reichenbach, M.; Stegmann, A.; Summerauer, L.; Wilken, F.; Fiener, P. TropSOC Database. Version 1.0. GFZ Data Services. https://doi.org/10.5880/fidgeo.2021.009, 2021.

Bukombe, B., Fiener, P., Alison M. Hoyt, A. M., Doetterl, S.: Controls on heterotrophic soil respiration and carbon cycling in geochemically distinct African tropical forest soils. SOIL Discuss. [pre-print], doi:10.5194/soil-2020-96, in review, 2021.

Reichenbach, M., Fiener, P., Garland, G., Griepentrog, M., Six, J. and Doetterl, S.: The role of geochemistry in organic carbon stabilization in tropical rainforest soils, Soil Discussion [pre-print], doi:10.5194/soil-2020-96, in review, 2021.

Wilken, F., Fiener, P., Ketterer, M., Meusburger, K., Muhindo, D. I., van Oost, K., and Doetterl, S.: Assessing soil erosion of forest and cropland sites in wet tropical Africa using $^{239+240}$Pu fallout radionuclides, SOIL Discuss. [preprint], https://doi.org/10.5194/soil-2020-95, in review, 2020.

**Referee comment 9.** Conclusion section – much of this is a stretch from the data. The conclusion section should be refocused on the data that is presented within the manuscript.

*Authors' response 9. Thank you for this comment. That was our ambition i.e., using the data to show what we learned. Considering your comment, the conclusion will be refocused on the results presented in the manuscript as follow:*

**5 Conclusions**

*Do patterns of microbial processes in deeply weathered tropical forest soils reflect the geochemical gradient of parent material under similar climatic conditions?*

We conclude that patterns of microbial biomass (MBC$_{Soil}$) and enzyme activity in tropical soils of contrasting geochemical origin are markedly distinct due to differences in resource availability. Similar patterns and magnitude of P limitation in these deeply weathered, undisturbed systems particularly in subsoils indicated that processes regulating P availability do not differ between soils of contrasting geochemical origin. Because of long-lasting efficient recycling of organic matter in humid tropics, none of the investigated regions and soil depths showed signs of N limitation for microbial processes.

*Are subsoil microbes in highly weathered tropical forest soils more C- and nutrient-limited than their topsoil counterparts?*

Less microbial C limitation in sub- compared to topsoils indicated that subsoil microbes can significantly participate in C cycling and limit C storage if high oxygen availability is prevalent. This is an observation that has so far not been highlighted enough in research on tropical biogeochemistry.

Furthermore, enhanced P limitation in sub- compared to topsoils indicated that the recycling of organic matter is the major P source as studied soils are depleted in rock-derived nutrients.

*What are important controls on soil microbial biomass, C and nutrient limitations?*

We conclude that microbial biomass C ($MBC_{Soil}$) is predominantly driven by organic matter. Control of organic matter over $MBC_{Soil}$ is strongly influenced by soil depth which could define the quality and quantity of organic matter as well as its interaction with other geochemical soil properties. In contrast, soil geochemistry and substrate quality are significant controls for $MBC_{SOC}$ particularly under conditions of less labile C. Strong controls of microbial C and P limitations could not be identified in this study referring to the large complexity of microbial resource acquisition.

**Referee comment 10.** Some more thoughts and questions: How much C was respired as CO2 over the incubation across treatments? (LINE 125) Did this vary statistically across treatments, and if so, how? Line 96. Spelling of schist Line 190. Spelling of bestNormalize Line 190.

***Authors' response 10.*** *As requested, we calculated the cumulative $CO_2$ release over the course of the incubation for different treatments (Fig. 1). Please note that respiration data in Bukombe et al. (2021, same special issue) is presented as weighted average of 12 CO2 sampling points during incubation. The weighing factor was defined as the number of incubation days represented by each CO2 collection point.*

[Figure]

*Figure 1. Cumulative $CO_2$ respiration during the 120-day incubation. Error bars are standard deviations (n=12). Letters compare geochemical regions per soil depth.*

*- Total $CO_2$ respired over the incubation was highest in the mafic region regardless of soil depth. Mixed sedimentary and felsic regions had statistically similar $CO_2$ release.*
*- We will correct the spelling of schist and bestNormalize in the manuscript.*

**Referee comment 11.** Which transformation was performed on which variables? Line 216. Define MBCDOC Line 314. EAA or EEA? Fig 1. The figures should be set up in more comprehensive way, with abbreviations defined and treatments given in the same order and with legends. Fig 5. Are these p values significantly different?

***Authors' response 11.*** *Thank you for these hints. As suggested, we will indicate in the manuscript the transformations used for each variable as follow:*

*In case of deviation from ANOVA assumptions (due to the natural variability of samples), we performed data transformation using the bestNormilize package in R. Log transformation was applied on EEA, orderNorm transformation was applied on vector characteristics, DOC and TDN, square root transformation was applied on MBCSoil, MBCSOC and MBCDOC (Peterson and Cavanaugh, 2019).*

*-As suggested, $MBC_{DOC}$ will be defined in the manuscript as follow: Increased $MBC_{SOC}$ and $MBC_{DOC}$*

*(microbial biomass C normalized to DOC) with soil depth were determined in the felsic region at pre-incubation.*

*-EAA was a mistake, the appropriate abbreviation (EEA for extracellular enzyme activity) will be provided throughout the discussion.*

*-For each figure we will define all abbreviations in the legend. Treatments will be presented in the same order throughout the result section.*

*- We compared all models (Fig. 5 or Table 2) and they are significantly different.*

**References**

Allison, S.D., Vitousek, P.M.: Responses of extracellular enzymes to simple and complex nutrient inputs. Soil Biol. Biochem., 37: 937–944, 2005.

Allison, S.D., Weintraub,M.N., Gartner,T.B., Waldrop, M.P.: Evolutionary-economic principles as regulators of soil enzyme production and ecosystem function, in Soil Enzymology, eds G. C. ShuklaandA.Varma(NewYork, NY:Springer-Verlag), 229–243.doi:10.1007/978-3-642-14225-3_12, 2011.

Deng H., Yu, Y-H., Sun, J-E., Zhang, J-B., Cai, Z-C., Guo, G-X., Zhong, W-H.: Parent materials have stronger effects than land use types on microbial biomass, activity and diversity in red soil in subtropical China, Pedobiologia, 58, 73-79, https://doi.org/10.1016/j.pedobi.2015.02.001, 2015.

Fierer N.: Embracing the unknown: disentangling the complexities of the soil microbiome, Nat. Reviews Microbiol., 15: 579–590, doi:10.1038/nrmicro.2017.87, 2017.

Fierer, N., Jackson, R.B.: The diversity and biogeography of soil bacterial communities; P Natl Acad Sci USA 103: 626–631, 2006.

Geisseler, D., Horwath, W.R., Joergensen, R.G., Ludwig, B.: Pathways of nitrogen utilization by soil microorganisms e A review, Soil Biol. Biochem., 2058–2067, doi:10.1016/j.soilbio.2010.08.021.

Hill, B.H., Elonen, C.M., Jica, T.M., Kolka, R.K., Lehto, L.L.P., Sebestyen, S.D., Siefert- Monson, L.R.: Ecoenzymatic stoichiometry and microbial processing of organic matter in northern bogs and fens reveals a common P-limitation between peatland types. Biogeochemistry 120:203–224, 2014.

Jing, X., Chen, X., Fang, J., Ji, C., Shen, H., Zheng, C., Zhu, B.: Soil microbial carbon and nutrient constraints are driven more by climate and soil physicochemical properties than by nutrient addition in forest ecosystems, Soil Biol. Biochem., 141, 107657. https://doi.org/10.1016/j.soilbio.2019.107657, 2020.

Liu, J., Chen, J., Chen, G., Guo, J., Li, Y.: Enzyme stoichiometry indicates the variation of microbial nutrient requirements at different soil depths in subtropical forests, PLoS ONE 15, e0220599, https://doi.org/10.1371/journal.pone.0220599, 2020.

Moorhead, D. L., Sinsabaugh, R. L., Hill, B. H., Weintrau, M. N.: Vector analysis of ecoenzyme activities reveal constraints on coupled C, N and P dynamics, Soil Biol. Biochem., 93, 1-7, http://dx.doi.org/10.1016/j.soilbio.2015.10.019, 2016.

Moorhead, D.L., Rinkes, Z.L., Sinsabaugh, R.L., Weintraub, M.N.: Dynamic relationships between microbial biomass, respiration, inorganic nutrients and enzyme activities: informing enzyme-based decomposition models, Front. Microbiol., 4, 223, https://doi.org/10.3389/fmicb.2013.00223, 2013.

Mori, T., Imai, N., Yokoyama, D., Kitayama, K.: Effects of nitrogen and phosphorus fertilization on the ratio of activities of carbon-acquiring to nitrogen-acquiring enzymes in a primary lowland tropical rainforest in Borneo, Malaysia. Soil Science and Plant Nutrition 1–4. https://doi.org/10.1080/00380768.2018.1498286, 2018.

Panchuk, K.: Igneous rocks, in physical geology, First University of Saskatchewan Edition by http://openpress.usask.ca/physicalgeology/, 2019.

Peterson, R.A, Cavanaugh, J.E.: Ordered quantile normalization: a semiparametric transformation built for the cross validation era. J. Appl. Stat.: 1-16, doi: 10.1080/02664763.2019.1630372, 2019.

Wang, C., Lu, X., Mori, T., Mao, Q., Zhou, K., Zhou, G., Nie, Y., Mo, J.: Responses of soil microbial community to continuous experimental nitrogen additions for 13 years in a nitrogen-rich tropical forest. Soil Biology and Biochemistry 121. https://doi.org/10.1016/j.soilbio.2018.03.009, 2018.

Wang, S., Mori, T., Mo, J., Zhang, W.: The responses of carbon- and nitrogen-acquiring enzymes to nitrogen and phosphorus additions in two plantations in southern China, J. Forest Res., 31, 319–1324 https://doi.org/10.1007/s11676-019-00905-0, 2019.